# LAION-COMP: UNLOCKING CONTROLLABLE AND COMPOSITIONAL GENERATION WITH STRUCTURAL ANNOTATIONS

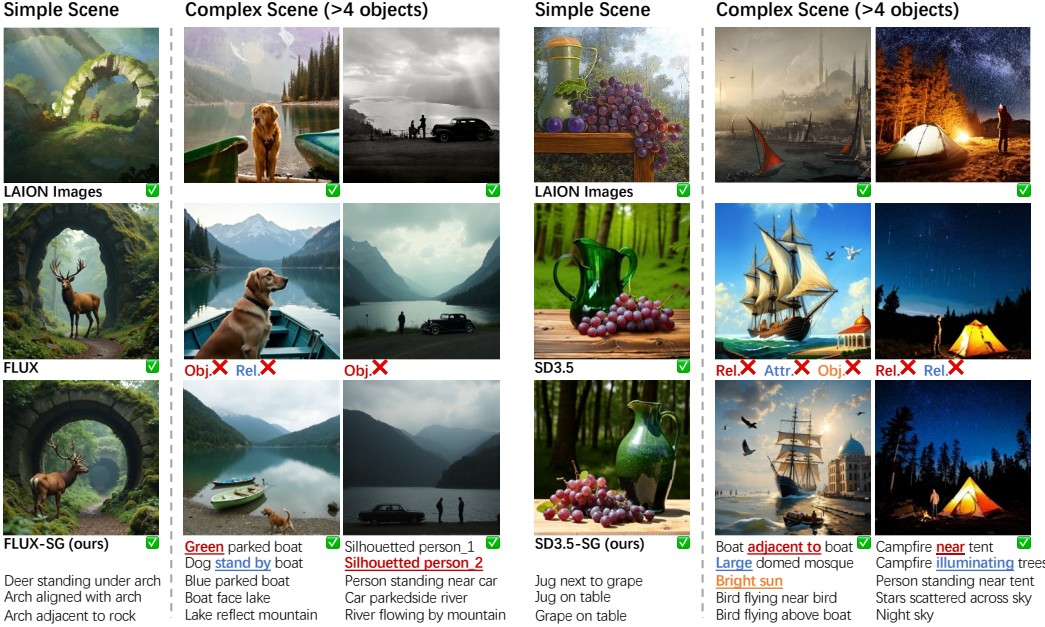

Figure 1: Images generated via prompt or translated structured annotations. We highlight inconsistent Obj(ect), Rel(ation), and Attr(ibute) in T2I Models. Models trained with our structured annotations perform significantly better than unstructured counterparts in complex scenes with >4 objects.

## ABSTRACT

Despite their success in generating high-quality images, text-to-image (T2I) models struggle to generate compositional scenes with multiple objects and their intricate relationships. We attribute this issue to limitations in existing datasets of image-text pairs, which lack precise inter-object relationship annotations with prompts only. To resolve this, we construct LAION-Comp, a large-scale dataset of 540K+ aesthetic images structurally annotated with detailed scene graphs explicitly encoding multiple objects, corresponding attributes, and intricate relations. The annotation pipeline employs a large vision-language model followed by partial human verification. Using LAION-Comp, we train 4 baseline models on diffusion and flow matching backbones augmented with a designed scene graph encoder. For proper evaluation, we introduce CompSGen Bench, a benchmark with 20,838 testing samples designed to systematically evaluate complex compositions. Experiments show that the 4 models trained on LAION-Comp outperform their original prompt-only counterparts and advanced scene-graph-based methods on both our new and existing compositional benchmarks. Furthermore, the learned structural conditioning naturally enables fine-grained, object-level image editing, demonstrating its potential as an effective editing interface. Our work validates the advantages of explicit structural annotation and contributes the community with a foundational resource to advance controllable and compositional image synthesis.

# 1 INTRODUCTION

Compositional image generation refers to the synthesis of scenes comprising multiple objects, their attributes, and intricate inter-object relations. As illustrated in fig. 1, conventional text-to-image (T2I) models Stability-AI (2024); Batifol et al. (2025) often falter when faced with such complexity. In contrast, generation frameworks guided by structured annotations demonstrate a superior capability in handling these scenarios accurately. We attribute this critical limitation not to model architecture, but to a fundamental deficiency in existing text-image datasets: a lack of explicit annotations for complex inter-object associations. Consequently, prior works that have primarily focused on architectural improvements have failed to address this underlying data-level issue.

To overcome this, we advocate for structural annotations, typically represented as scene graphs (SGs). An SG consists of nodes, representing objects and their attributes, and edges, depicting the relations between objects. In contrast to the inherently sequential and often ambiguous nature of text descriptions, SGs provide a compact, structured, and explicit paradigm for describing complex scenes, thereby enhancing annotation efficiency. Crucially, SGs enable the precise specification of specific objects associated attributes and their relations—a capability that is critical for both generating complex scenes and enabling fine-grained image editing. However, progress in this direction is hindered by a critical gap in data resources: existing scene graph datasets, such as COCO-Stuff (Caesar et al., 2018) and Visual Genome (Krishna et al., 2017), are limited in scale and diversity of annotation, while large-scale datasets consist almost exclusively of unstructured text annotations.

In this work, we aim to establish a more robust structural data foundation for compositional image generation while unlocking the potential of structured data for image editing tasks. Specifically, we construct LAION-Comp, a large-scale dataset built as a significant extension of LAION-Aesthetics V2 (6.5+) (Schuhmann et al., 2022) with high-quality, high-complexity structural annotations. Therefore, our LAION-Comp better encapsulates the semantic structure of complex scenes, supporting improved generation for intricate scenarios. The superiority of LAION-Comp in complex scene generation is validated in experiments with multiple metrics on semantic consistency.

Leveraging LAION-Comp, we train existing state-of-the-art models and propose a new suite of baseline models to comprehensively validate the effectiveness of structural annotations for compositional generation. Our baselines are built upon diffusion (Rombach et al., 2022; Podell et al., 2023) and flow matching (Stability-AI, 2024; Batifol et al., 2025) backbones. We design and train an auxiliary scene graph encoder that employs a Graph Neural Network (GNN) (Scarselli et al., 2008b) to effectively process the structural information in SGs and produce optimized embeddings. These embeddings are then integrated into the generative backbones, significantly enhancing models' capability to synthesize high-quality, complex images.

For a targeted and rigorous evaluation, we establish CompSGen Bench, a new benchmark specifically designed for complex scene generation. With this benchmark we evaluate leading T2I and SG-to-Image (SG2IM) models alongside our proposed baselines, comparing performance when trained on COCO-Stuff, Visual Genome, and our LAION-Comp. Both quantitative and qualitative results unequivocally demonstrate that models trained on LAION-Comp consistently and significantly outperform their counterparts. These findings lead us to conclude that the high-quality, large-scale structural annotations in LAION-Comp are crucial for advancing complex scene generation.

Furthermore, the structured nature of SGs naturally facilitates fine-grained, object-level image editing, as it allows users to perform intuitive and precise modifications directly on the graph structure. Building on this potential, we develop a training-free image editing framework based on an RF inversion strategy (Rout et al., 2025). Our qualitative and quantitative experiments demonstrate the remarkable effectiveness and controllability that structural annotations bring to image editing. Due to space limitation, the proposed editing framework is introduced in Sec. A.1.

In summary, our work represents a significant step toward scaling structurally complex annotations to high-quality, large-scale datasets, enabling broader scene synthesis and editing. Our contributions are as follows. (1) We introduce LAION-Comp, a new, large-scale dataset for compositional generation. It features high-quality structural annotations with multiple objects, attributes, and intricate relations, enhancing a model's ability to generate complex and high-fidelity images. (2) We fine-tune a new suite of foundation models based on diffusion and flow-matching backbones, demonstrating superior performance in complex scene generation. Furthermore, we propose a training-free, SG-

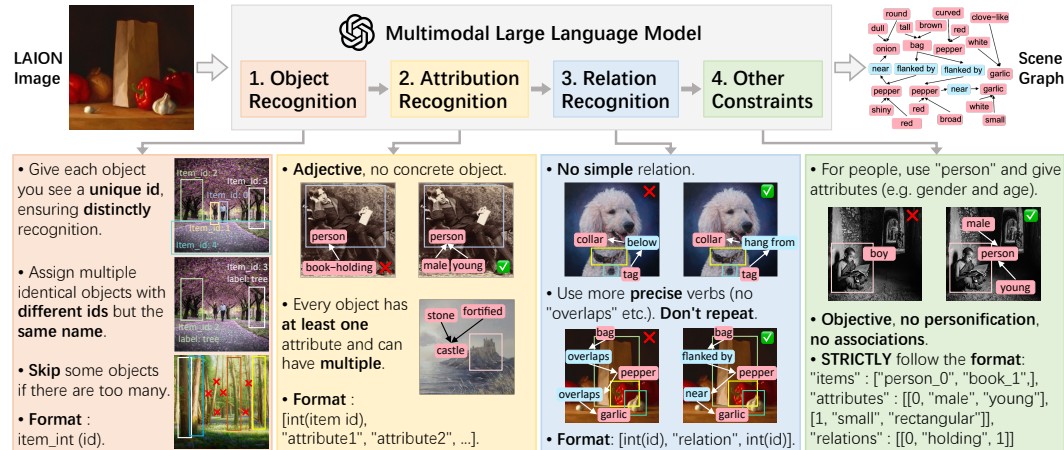

Figure 2: The construction pipeline of LAION-Comp dataset. 1) Objects in the image are identified and assigned unique ID's. 2) The attributes must be abstract adjectives and should not include specific objects. Each object may have one or more attributes. 3) The relations between objects should be as specific as possible with minimal repetition and avoid being labeled simple relations. 4) People are labeled as "person" with attributes such as gender and age. Anthropomorphism or associations are avoid and objective description of what is observed in the image are prefered.

based image editing framework, highlighting the powerful editing potential of structural annotations. (3) We establish CompSGen Bench, a dedicated benchmark to evaluae complex scene generation. Through extensive experiments on this benchmark, we validate the significant advantages of our dataset and the effectiveness of our proposed models. Our annotations with the associated processing code, the foundation models and the benchmark protocol will be publicly available.

## 2 RELATED WORK

**Compositional Image Generation.** Text-to-image generation (Saharia et al., 2022; Ramesh et al., 2022; Dhariwal & Nichol, 2021; Chen et al., 2024a; Tewel et al., 2024; Zhou et al., 2023; Li et al., 2024; Podell et al., 2023) has advanced significantly, particularly through diffusion models (Ho et al., 2020; Rombach et al., 2022). However, the sequential format of the textual data imposes limitations in handling compositional images with multiple objects and relations(Yang et al., 2024; Lian et al., 2023; Zhang et al., 2024b).

Methods enhance controllability via custom losses and attention maps, such as Universal Guidance (Bansal et al., 2023), BoxDiff (Xie et al., 2023), and RealCompo (Zhang et al., 2024a). Other approaches exploit spatial conditions (e.g., GLIGEN (Li et al., 2023), Ranni (Feng et al., 2024)) or LLM-assisted layouts (Feng et al., 2023b; Lian et al., 2023; Zhang et al., 2023a; Wu et al., 2024b), typically relying on precise inputs or incurring high training costs. All of these mainly focus on model improvement, failing fundamentally to address the limitations imposed from the dataset.

**Image Generation from Scene Graphs** (SG2IM) (Johnson et al., 2018; Krishna et al., 2017) involves creating images based on structured representations of scenes, where objects and their relationships are explicitly defined as a graph (Xu et al., 2017). Modern SG2IM models align scene graphs directly to images for better handling of content generation (Feng et al., 2023a; Wang et al., 2024a; Zhang et al., 2023b), with SG-Adapter fine-tuning Stable Diffusion (SD) via attention (Shen et al., 2024), SGDiff pre-trains an SG encoder combined with SD (Yang et al., 2022), and R3CD using transformers for abstract interactions (Liu & Liu, 2024). Refinements include knowledge consensus for semantic disentanglement (Wu et al., 2023b), cross-attention for object consistency (Zhang et al., 2023b), and masked auto-encoders for grounding in SGG-IG (Wang et al., 2025). These approaches enhance semantic capacity beyond text-only conditions, yet remain constrained by the limited scale and quality of existing SG datasets.

**Large-Scale Image-Text Datasets and Benchmarks.** Previous datasets, such as MS-COCO (Lin et al., 2014), Visual Genome (Krishna et al., 2017), and ImageNet (Deng et al., 2009), are limited

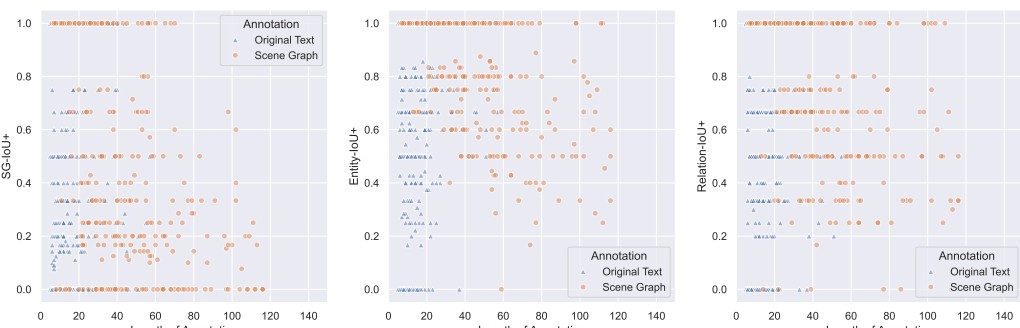

Figure 3: The annotation length and accuracy characteristics of LAION-Comp compared to the LAION-Aesthetics. Compared to the original text annotation, our labeled scene graphs, although as more compact forms, have longer lengths and higher accuracies concentrated in high-scoring areas. Our LAION-Comp annotation accurately reflects image information and contains richer semantics.

| Annotation | # Objects (w/o Proper Noun) | Length | SG-IoU+$^{\uparrow}$ | Ent.-IoU+$^{\uparrow}$ | Rel.-IoU+$^{\uparrow}$ |
|---|---|---|---|---|---|
| LAION Caption | 5.33±3.94 (2.02±3.01) | 19.0±19.7 | 0.306 | 0.631 | 0.557 |
| LAION-Comp | 6.39±4.17 | 32.2±20.3 | **0.422** | **0.810** | **0.749** |

Table 1: The number of objects and length per sample, and the average accuracy for 300 samples across different annotation types.

in scale due to the considerable costs associated with manual annotation. To mitigate the limitation, several studies have explored automatic annotation, as exemplified by CC12M (Changpinyo et al., 2021), SPRIGHT (Chatterjee et al., 2025), and LAION-5B (Schuhmann et al., 2022). LAION-Aesthetics is curated for high visual quality and intended to support image generation. However, it does not ensure textual descriptions that accurately reflect image content. Thus We enhance LAION-Aesthetics with structured annotations for high-quality compositional generation, adding attributes beyond objects, in contrast to contemporaneous effort (Chen et al., 2024b).

Benchmarks assess T2I comprehensively: T2I-CompBench for 6K prompts (Huang et al., 2023), HRS-Bench for 13 skills (Bakr et al., 2023), HEIM for 12 dimensions (Lee et al., 2023b), VISOR for spatial relations (Gokhale et al., 2023), and HPS v2 for human preferences (Wu et al., 2023a). Recent frameworks add flexibility, like ConceptMix for controllable difficulty (Wu et al., 2024a), INQUIRE for expert queries (Vendrow et al., 2024), and GenEval for object-focused metrics (Ghosh et al., 2023). These benchmarks only focus on text-based image generation. To fill the gap in this domain, we are the first to propose a compositional generation benchmark based on scene graphs.

## 3 DATASET AND BENCHMARK

A large-scale, high-quality dataset is essential for learning compositional image generation. However, existing large-scale T2I datasets, such as LAION (Schuhmann et al., 2022), describe information beyond the images (as illustrated in fig. 5), misleading the generation. In contrast, SG datasets tend to focus more specifically on the actual content within images, namely the objects and relations. Nonetheless, current SG datasets, such as COCO and VG, are relatively small in scale and have limited object and relationship types, making them insufficient for compositional image generation.

To address this, we propose LAION-Comp, a large-scale, high-quality, open-vocabulary SG dataset and Complex Scene Generation Benchmark (CompSGen Bench) to evaluate models' performance .

### 3.1 DATASET CONSTRUCTION

Our LAION-Comp dataset is built on high-quality images in LAION-Aesthetic V2 (6.5+) (Schuhmann et al., 2022) with automated annotation performed using GPT-4o (OpenAI et al., 2024). LAION-Aesthetics V2 (6.5+) is a subset of LAION-5B (Schuhmann et al., 2022), comprising

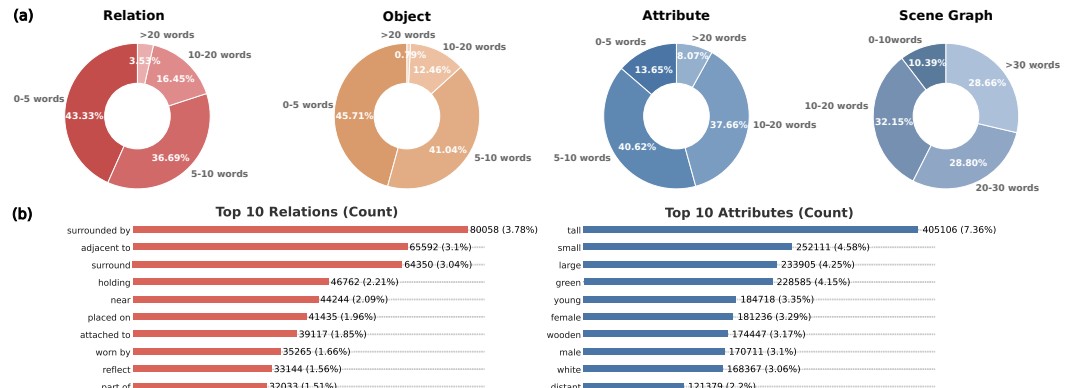

Figure 4: The annotation distribution of LAION-Comp. (a) The length of scene graphs lies in a wide range. Our annotation provides more specific information compared to single-word descriptions, while avoiding the inefficiency in model learning caused by lengthy annotations. (b) The top 10 relations and attributes represent a small percentage of the total distribution, indicating LAION-Comp covers a highly diverse range of annotations, showcasing its large scale and open vocabulary.

625,000 image-text pairs with predicted aesthetic scores over 6.5, curated using the LAION-Aesthetics Predictor V2 model. During our construction, only 540,005 images are available.

Through prompt engineering, we devised a set of specific requirements for scene graph annotations to ensure comprehensiveness, systematic structure, and precision in the annotation results. Figure 2 illustrates the detailed construction pipeline of LAION-Comp. Each component plays a crucial role in achieving high-quality automated annotation.

First, as scene graphs typically contain multiple objects and their relations, the prompt requires "identification of as many objects, attributes, and their relations within the image as possible". This design encourages that all objects and interactions in a scene are annotated. Each object is assigned a unique ID, even for multiple objects of the same type, ensuring that the entirety of the scene's structure and hierarchy is accurately represented.

Second, the attribute section mandates that each object must have at least one abstract adjective attribute, while avoiding the use of other objects as attributes. This design is especially important in complex scenes as it helps differentiate objects' appearance, state, and characteristics from the background and other elements, maintaining consistency and clarity in annotations. By avoiding the confused annotation between specific objects and abstract attributes, the annotations become more interpretable and generalizable.

In the relation section, we specify the use of concrete verbs to describe relations between objects rather than relying solely on spatial orientation. This is because relations are often more critical in scene graphs than mere spatial information. By using precise verbs like "standing on" or "holding", we capture dynamic interactions within the scene, which is essential for complex scene generation.

Leveraging these prompts with the multimodal large language model GPT-4o, we generate annotations representing scene graphs. To investigate the reliability of the annotations, we conduct a partial human verification. Results show the annotations achieve high accuracies of 98.8% for objects, 97.5% for attributes, and 95.7% for relations (Sec. A.5).

## 3.2 LAION-COMP DATASET

By performing the construction strategy, we develop LAION-Comp, a large-scale, high-quality dataset containing 540,005 SG-image pairs annotated with objects, attributes, and relationships. This dataset is divided into a training set of 480,005 samples, a validation set of 10,000 samples, and a test set of 50,000 samples. We present statistics comparing the original LAION-Aesthetics text-to-image dataset with our LAION-Comp dataset as follows.

In table 1, in the original LAION-Aesthetics caption, the average number of objects per sample is 5.33, with 38% of these being proper nouns that offer limited guidance during model training. For

our SG annotations, the average number of objects per sample increases to 6.39, excluding abstract proper nouns and focusing on specific nouns that reflect true semantic relationships. LAION-Comp contains 20% more object information than the original LAION-Aesthetics dataset, and this advantage increases to 216% when excluding proper nouns. We also calculated the relationship between length and accuracy for different annotations. The annotation length for text is defined as the number of tokens in the prompt, while for SG as the total number of nodes and edges. We leverage SG-IoU+, Entity-IoU+, and Relation-IoU+ introduced in Sec. A.2 to measure annotation accuracy.

The average annotation length for original captions and our scene graphs is 19.0 and 32.2, respectively, with SG achieving higher accuracy across all three metrics. Figure 3 visualizes the length and accuracy of samples for both annotation types. Note that a scene graph is a more structured and compact form of annotation compared to text. Even so, the annotated SG length is still significantly longer than sparse text, and its accuracy is also much higher. This demonstrates that our LAION-Comp dataset contains richer, more nuanced, and precise semantic features, enhancing the trained model performance and fundamentally addressing the challenges of generating complex scenes.

Furthermore, we analyze the length distribution of scene graphs in LAION-Comp in fig. 4 (a). Most objects are described by 0-5 (45.72%) or 5-10 (41.04%) words, with a smaller proportion described by 10-20 (12.46%) words or $\geq$ 20 (0.79%) words. This range is reasonable, offering a more precise expression than a single word while avoiding excessive length that could hinder model learning efficiency. In terms of the overall scene graph, the proportions of word counts in the ranges 0-10, 10-20, 20-30, and $\geq$ 30 are 10.39%, 32.15%, 28.80%, and 28.66%, respectively. These statistics reflect the richness, detail, and flexibility of annotations in LAION-Comp.

Figure 4 (b) presents the top 10 most frequent relations and attributes in LAION-Comp. The most frequent relation is "surrounded by", occurring 80,058 times and accounting for 3.78% of all relations. The 1st common attribute is "tall" (7.36%), while the 2nd common is "small" (only 4.58%). The 10th relation and attribute each make up only 1.51% and 2.2%. These data indicate the annotations in LAION-Comp are highly diverse and broadly covered, as even the most frequently used descriptors represent only a small percentage.

To highlight the semantic richness and diversity of LAION-Comp, we conduct a comparative analysis with the widely used VG (Krishna et al., 2017), focusing on the distribution of relation types. Specifically, we categorize relations into spatial (e.g., "on", "under", "next to") and non-spatial (e.g., "holding", "wearing", "playing") types, which reflect different levels of semantic complexity. Quantitative analysis highlights a clear distributional difference. In LAION-Comp, non-spatial relations dominate (77.48%), whereas spatial relations account for only 22.52%. Conversely, VG is spatially skewed, with 58.02% versus 41.98%. LAION-Comp captures more abstract, functional, and interaction-based semantics, moving beyond the predominantly geometric or locational focus of VG. Such enrichment is crucial for compositional and controllable image generation, providing a more challenging and realistic benchmark for scene understanding, as also reflected in T2I-CompBench (Huang et al., 2023) and MMRel (Nie et al., 2024), where models exhibit greater difficulty with complex non-spatial semantics than with spatial configurations.

### 3.3 COMPLEX SCENE GENERATION BENCHMARK

To evaluate model performance on compositional image generation, we propose Complex Scene Generation Benchmark (CompSGen Bench). From the 50,000-image test set, we select samples with over four relations as complex scenes, and get a total of 20,838 samples. We calculate FID (Lee et al., 2023a) , CLIP score (Radford et al., 2021), and three accuracy metrics (Shen et al., 2024) to assess models' performance. FID measures the overall quality of generated images, while the CLIP score calculates the similarity between the generated and ground truth images. The complex scene evaluation consists of three metrics: SG-IoU, Entity-IoU, and Relation-IoU. They represent the overlap between the generated images and the real annotations in terms of scene graphs, objects, and relations, respectively. Sec. 5.1 shows the test results for different models on CompSGen Bench.

## 4 FOUNDATION MODELS

As the complexity of the prompt increases, the generated image becomes more difficult to control (fig. 1). We introduce foundation models to address the challenges of compositional image gener-

ation in T2I task. Our models are built on advanced diffusion (Podell et al., 2023; Rombach et al., 2022) and flow matching (Stability-AI, 2024; Batifol et al., 2025) backbones, incorporating structural information via graph neural networks (GNN) (Scarselli et al., 2008b).

A scene graph consists of multiple triples and single objects. Our baseline initializes each triple and single object separately using the CLIP text encoder $E_T(\cdot)$. For single objects, the initialization result from CLIP serves as the final representation, denoted as $\mathbf{e}_s$. For SG triples, each of them is encoded by CLIP to yield a corresponding triple embedding $\mathbf{e}_t = E_T(triple^{sg})$. Our SG encoder extracts object and relation embeddings as the nodes and edges and inputs them into the GNN to optimize the SG embedding. More calculation details can be found in Sec. A.9.3.

If a relation contains multiple words, each word contributes an edge connecting the nodes of the two related objects. Attributes are treated as separate nodes connected to their respective objects. After processing with the GNN, we obtain a refined triple embedding, denoted as $\mathbf{e}_r$.

To stabilize the training, we introduce a learnable scaling factor $\alpha$ to control the strength of the refined embedding. $\alpha$ is initialized as zero and updated throughout training. Finally, all triple embeddings are concatenated with single-object embeddings to form the SG embedding $\mathbf{e}_{sg}$, which is fed into diffusion- or flow-matching-based backbones for compositional semantic learning.

$$\mathbf{e}_{sg} = f(sg) = \text{concat}(\mathbf{e}_t + \alpha\mathbf{e}_r, \mathbf{e}_s) \tag{1}$$

Taking flow-matching-based backbones as an example, given a clean image latent $x_0$ and Gaussian noise $\epsilon$, the SG encoder is trained with:

$$\mathcal{L} = \mathbb{E}_{x_0, \epsilon, t, sg} \left[ \| v_\theta(z_t, t, f(sg)) - (\epsilon - x_0) \|_2^2 \right], \tag{2}$$

where $z_t$ is the rectified flow trajectory, $t \in [0, 1]$, and $f(sg)$ denotes the SG embedding. We train the parameters of SG encoder to minimize the gap between the predicted and ground-truth vector field, which are defined as $v_\theta$ and $u_t(z|\epsilon) = \epsilon - x_0$. This objective is shared across SD3.5-SG and FLUX-SG, while the integration strategy of SG embedding differs. Sec. A.9.4 provides a more detailed derivation of this process and Sec. A.9.3 elucidates the theoretical principles of the diffusion-based baselines. Our scene graph encoder is fine-tuned to align with the generative architectures of these models, leading to enhanced synthesis performance. To enhance user-friendliness, we design an automated pipeline that supports flexible, dual-modality inputs: free-form text and structured SGs (Sec. A.9.5). And the editing framework based on the foundational model is introduced in Sec. A.1.

## 5 EXPERIMENTS

Our trained models for compositional generation are comprehensively evaluated against several strong baselines (Podell et al., 2023; Shen et al., 2024; Yang et al., 2022) on the CompSGen Bench, COCO-Stuff, and Visual Genome datasets. In addition, we present experimental results on SG-based image editing in Sec. A.1.2 and Sec. A.1.3. We also conduct a quantitative analysis (Sec. 3.2) and a user study (Sec. A.3) to verify the effectiveness and strong correlation with human perception of structured annotations. Further details regarding the experimental setup are available in Sec. A.2.

### 5.1 COMPOSITIONAL IMAGE GENERATION

**Qualitative Results.** Figure 5 displays 1024×1024 images generated on LAION-Comp. Each row shows the original caption, the scene graph, the GT image, and images generated by different models. The corresponding elements in the SG and images are highlighted in matching colors.

For fairness, we compare our SDXL-SG with existing diffusion-based SG2IM models, while the results of FLUX-SG are provided at the end. SDXL-SG and FLUX-SG can generate scenes with more accurate objects and relations, even for complex scenarios. For instance, in the first row, where the relationship is "male person painting female person", both (a) and (b) fail to generate "painting", and (c) generates two females, whereas SDXL-SG accurately and qualitatively generate the provided relations. Figures (f)-(t) illustrate more examples where ours outperform existing baselines.

Additionally, existing T2I and SG2IM models more frequently generate incorrectly in (f). Other errors include erroneous number of generated objects such as bag in the green box in (k), person in the blue box in (p) and (q) or attribute errors such as bag in the green box in (l). Conversely, SDXL-SG and FLUX-SG demonstrate robustness against these failure modes.

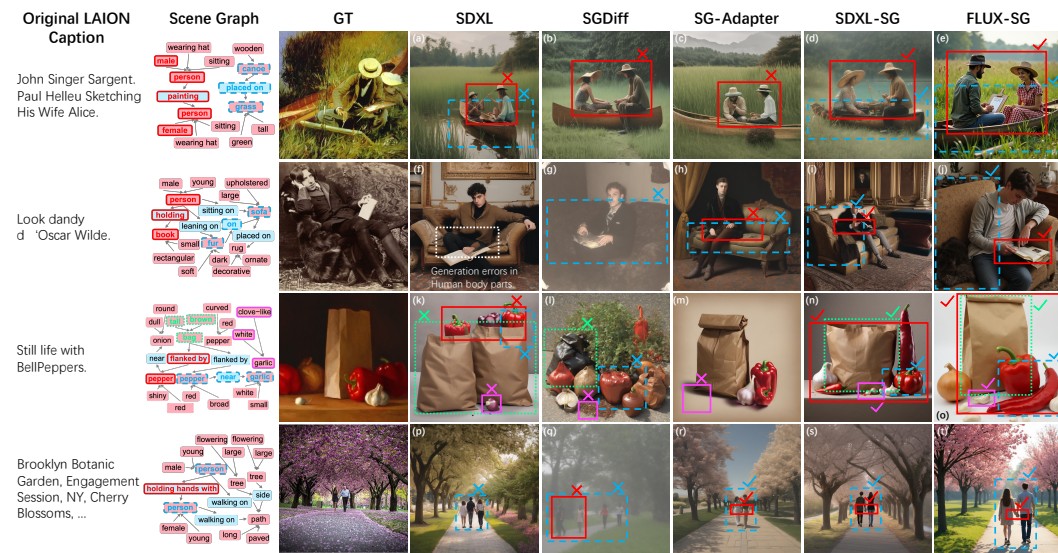

Figure 5: Visual comparison on LAION-Comp. The compared methods include T2I model (SDXL (Podell et al., 2023)) and SG2IM models (SGDiff (Yang et al., 2022) and SG-Adapter (Shen et al., 2024)). The 1st column shows the original caption from LAION-Aesthetics. The 2nd column displays the SG from our LAION-Comp. The last 6 columns show GT images and images generated by different models. Objects or relations are highlighted with the same color in scene graphs and generated images to show SDXL-SG and FLUX-SG successfully capture complex scenes.

| Type | Method | Dataset | FID↓ | SG-IoU↑ | Ent.-IoU↑ | Rel.-IoU↑ |
|------|--------|---------|------|---------|-----------|-----------|
| T2I | SDXL | LAION | **19.3** | 0.371 | 0.813 | 0.780 |
| | SD3.5-Medium | LAION | 24.6 | 0.541 | 0.854 | 0.831 |
| | FLUX.1-Dev | LAION | 26.2 | 0.544 | 0.885 | 0.842 |
| SG2IM | SGDiff w/o bbox | COCO | 47.8 | 0.435 | 0.841 | 0.816 |
| | | Visual Genome | 35.2 | 0.529 | 0.801 | 0.795 |
| | | LAION-Comp | 32.2 | 0.531 | 0.855 | 0.830 |
| | SG-Adapter | COCO | 34.9 | 0.485 | 0.840 | 0.833 |
| | | Visual Genome | 39.5 | 0.515 | 0.803 | 0.782 |
| | | LAION-Comp | 31.3 | 0.538 | 0.866 | 0.852 |
| | SDXL-SG (Ours) | COCO | 30.0 | 0.497 | 0.842 | 0.833 |
| | | Visual Genome | 21.9 | 0.546 | 0.813 | 0.800 |
| | | LAION-Comp | 20.1 | 0.558 | 0.884 | 0.856 |
| | SD3.5-SG (Ours) | LAION-Comp | 20.8 | 0.578 | **0.897** | 0.859 |
| | FLUX-SG (Ours) | LAION-Comp | 24.7 | **0.583** | 0.893 | **0.859** |

Table 2: Quantitative results. The first and second best is in **bold** and underlined.

**Quantitative Results.** We compared results of both T2I and SG2IM models trained on different datasets. The original SGDiff (Yang et al., 2022) introduces bounding box as auxiliary data during training. For fair comparison, we train SGDiff without bounding box with the official implementation. We used FID to evaluate the quality of generated images. Fine-tuning pre-trained T2I models inevitably increases FID scores (Ruiz et al., 2023; Shen et al., 2024; Wang et al., 2024c). We also measure SG-IoU, Entity-IoU, and Relation-IoU (Shen et al., 2024).

As demonstrated in table 2, our baseline achieves the best performance among all candidates in both image quality and accuracy. Notably, the SG-IoU of T2I model is significantly lower than that of SG2IM models, indicating that text provides far less control in the image generation process compared to structured annotations. This highlights the necessity of constructing a large-scale, high-quality structured annotation dataset. Furthermore, for the same model, results trained on LAION-

| Type | Method | FID↓ | CLIP↑ | SG-IoU↑ | Ent.-IoU↑ | Rel.-IoU↑ |
|---|---|---|---|---|---|---|
| T2I | SD1.5 | 60.4 | 0.654 | 0.170 | 0.604 | 0.511 |
| | SDXL | **25.2** | 0.700 | 0.226 | 0.753 | 0.658 |
| SG2IM | SGDiff | 35.8 | 0.690 | 0.304 | 0.787 | 0.698 |
| | SG-Adapter | 27.8 | 0.681 | 0.314 | 0.771 | 0.693 |
| | SD1.5-SG* | 56.3 | 0.653 | 0.179 | 0.614 | 0.530 |
| | SDXL-SG* | 26.7 | 0.698 | 0.340 | 0.792 | 0.703 |
| | SD3.5-SG* | 28.5 | 0.702 | **0.345** | 0.840 | 0.738 |
| | FLUX-SG* | 29.0 | **0.707** | 0.338 | **0.851** | **0.776** |

Table 3: T2I and SG2IM results on the CompSGen Benchmark. * denotes ours. The best is in **bold**, and the second best is underlined.

Comp consistently outperformed those trained on COCO and VG. This suggests that our LAION-Comp is more effective than previous SG-image datasets due to its higher annotation quality.

Additionally, we evaluate the complex scene generation capability of advanced T2I and SG2IM models on the CompSGen Bench (Sec. 3.3). As shown in table 3, our baseline outperforms existing models in terms of image quality, similarity to GT images, and content accuracy. Compared to SDXL, the FID of SDXL-SG does not increase significantly after fine-tuning—a process that typically elevates FID. However, SDXL-SG substantially outperforms SDXL on accuracy metrics, including SG-IoU, Entity-IoU, and Relation-IoU. Beyond the SDXL backbone, we also perform evaluations using SD1.5 and the flow-matching-based SD3.5-SG and FLUX-SG, which achieve further performance gains, indicating the effectiveness and adaptability of our dataset and method.

We further compute CLIP scores on COCO, which are 0.630 for SDXL and 0.635 for SDXL-SG. Although the test set of CompSGen Bench is more complex, the models achieve even higher scores, corroborating the high quality of LAION-Comp. Moreover, we conduct evaluations on T2I-CompBench (Huang et al., 2023), with details provided in Sec. A.6, which demonstrate the superiority of our dataset and baseline model.

## 5.2 ABLATION STUDY

We conduct ablation studies to demonstrate the positive impact of LAION-Comp. We train SDXL-SG variants on 10%, 20%, 50%, and 100% samples of LAION-Comp. The total training iterations remain constant across all settings for fairness. As the sample size increases, the model's capability to generate compositional images improves significantly (table 4). Notably, in the 10% LAION-Comp ablation, where the data volume is smaller than that of VG, the model's FID and Entity-IoU scores still outperform the

| Method | Prop. | FID↓ | SG-IoU↑ | Ent.-IoU↑ | Rel.-IoU↑ |
|---|---|---|---|---|---|
| SG-Adapter | 10% | 31.6 | 0.522 | 0.794 | 0.790 |
| | 20% | 24.3 | 0.524 | 0.804 | 0.793 |
| | 50% | 22.9 | 0.535 | 0.800 | 0.796 |
| | 100% | 21.9 | 0.546 | 0.813 | 0.800 |
| SDXL-SG | 10% | 27.3 | 0.530 | 0.874 | 0.837 |
| | 20% | 24.5 | 0.533 | 0.877 | 0.838 |
| | 50% | 22.2 | 0.547 | 0.876 | 0.849 |
| | 100% | **20.1** | **0.558** | **0.884** | **0.856** |

Table 4: Results of ablation. Prop. denotes data proportion.

results trained on VG, with other scores remaining roughly comparable (table 2). LAION-Comp not only provides a data volume advantage but also features higher quality in images and annotations, which enhances training efficiency and improves performance in compositional image generation.

## 6 CONCLUSION

We introduce LAION-Comp, a large-scale dataset with detailed structural annotations for compositional generation, addressing the core problem of unstructured training data. Models trained on LAION-Comp demonstrate improved fidelity and compositional accuracy on our CompSGen Bench and existing benchmarks, outperforming present methods. Our work validates that large-scale, high-quality structural annotations are crucial for advancing controllable image synthesis and provides a foundational resource to the community for future research.

**Reproducibility Statement**

To ensure the reproducibility of our research, we provide detailed descriptions of our methods. The guidelines for our dataset construction process are detailed in Sec. 3.1. We describe our foundation models for compositional generation in Sec. 4 and Sec. A.9, and the specifics of SG-based image editing in Sec. A.1. Our experimental setup is described in Sec. A.2. Furthermore, we have made the corresponding code for each model available in the supplementary material.

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

# A APPENDIX

## A.1 WORLD-KNOWLEDGE-AWARE SG-BASED IMAGE EDITING

We utilize SGs as an editing interface for fine-grained, object-level image editing. In contrast to verbose text prompts, this structured approach allows for direct and precise manipulation of objects and their relations, significantly improving the intuitiveness, efficiency, and controllability of the editing process. The core of our framework is a training-free SG-consistent RF-inversion strategy. Initially, the original scene graph ($sg$) and its edited version ($sg'$) are encoded into embeddings $\mathbf{e}_{sg}$ and $\mathbf{e}'_{sg}$, respectively. During inversion, the original embedding $\mathbf{e}_{sg}$ conditions the process to yield an aligned initial latent variable, enhancing controllability compared to conventional null-prompt methods (Esser et al., 2024). Subsequently, the editing stage uses the modified embedding $\mathbf{e}'_{sg}$ as the new conditioning signal to synthesize an image that precisely reflects the user's modifications.

For flexible editing, we introduce a world-knowledge-aware image editing agent (fig. 6), which integrates a user intent parser. This component leverages the reasoning capabilities of LLMs to systematically decompose free-form instructions into a sequence of structured scene graph modifications targeting objects, attributes, and relations, ensuring the proposed edits adhere to real-world physical laws and common sense.

### A.1.1 DETAILS ON SG-BASED IMAGE EDITING

In addition to compositional image generation, we explore the potential of structural conditions for fine-grained, object-level image editing. Our proposed framework utilizes scene graphs as its editing interface and supports a broad spectrum of editing operations, including object addition, replacement, deletion, and relationship modification. Unlike unstructured text conditions, which often require verbose descriptions for object localization, a structural interface enables users to perform intuitive and precise modifications directly on the scene graph. This direct manipulation greatly improves the convenience, efficiency, and controllability of image editing.

To bridge user intent and graph-based manipulation, we introduce a world-knowledge-aware image editing agent ( Figure 6). It allows users to either (i) directly edit the SG structure, or (ii) provide free-form natural language instructions. In the latter case, a language parsing module, powered by large language models, interprets user commands into corresponding graph operations (modifications of objects, relations or attributes) while enforcing physical plausibility via built-in commonsense and

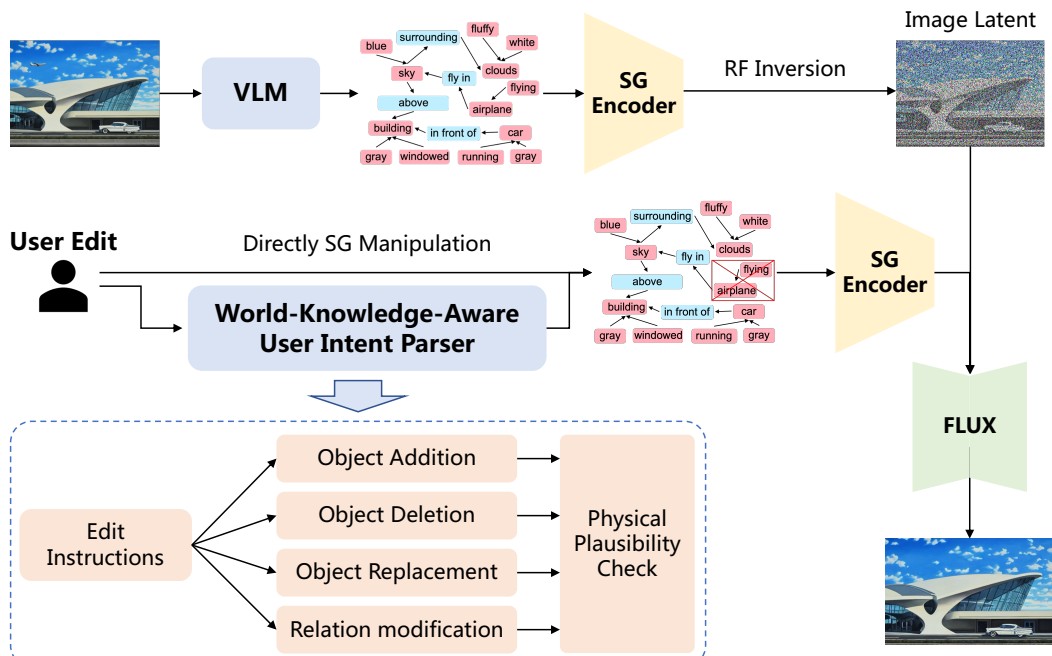

Figure 6: The pipeline of world-knowledge-aware image editing.

| Method | EC$^\uparrow$ | RA$^\uparrow$ | IQ$^\uparrow$ |
|---|---|---|---|
| InstructP2P | 0.616 | 0.679 | 0.564 |
| SGEdit | 0.717 | 0.735 | 0.562 |
| RF Inversion | 0.894 | 0.898 | 0.871 |
| FLUX-SG | **0.899** | **0.915** | **0.902** |

Table 5: Quantitative results of SG-based editing. EC, RA, and IQ denote the win rates of element composition, relational alignment, and image quality.

world knowledge constraints. The modified SG is then encoded by a pre-trained SG encoder, yielding a semantic embedding $\mathbf{e}'_{sg}$ that conditions the subsequent editing process.

At the core of our framework lies an SG-consistent RF-inversion strategy. Specifically, given the original scene graph $sg$ and the user-edited version $sg'$, we encode them into embeddings $\mathbf{e}_{sg}$ and $\mathbf{e}'_{sg}$ by our pre-trained SG encoder, respectively. During the inversion stage, the original SG embedding $\mathbf{e}_{sg}$ is introduced as a condition to yield a latent variable that is used to initialize the editing process. We compute the vector field as $v_t(\mathbf{x}_t) = -g(\mathbf{x}_t, 1 - t, \mathbf{e}_{sg}; \varphi)$, where $g$ is the pre-trained FLUX model parameterized by $\varphi$. Unlike conventional RF inversion (Rout et al., 2025), which often operate with null prompts condition, our FLUX-SG inversion enforces alignment between the image latent space and SG condition, leading to more controllable editing. In the subsequent editing stage, we use the modified SG embedding $\mathbf{e}'_{sg}$ to replace $\mathbf{e}_{sg}$ as the conditioning signal for vector field computation, ensuring that the synthesized image accurately reflects user-specified modifications.

### A.1.2 QUANTITATIVE RESULTS ON IMAGE EDITING

We evaluate the effectiveness of our model on the SG-based image editing task across three dimensions: element composition (EC), relationship alignment (RA), and image quality (IQ). Following prior work (Zhang et al., 2024c), we randomly sample 30 real images from the LAION-Aesthetic dataset and perform four types of editing operations: object addition, replacement, deletion, and relationship modification, resulting in 120 editing scenarios per model. We conduct a comprehensive comparison against SG-based (SGEdit (Zhang et al., 2024c)) and a text-based (InstructP2P (Brooks et al., 2023), RF Inversion (Rout et al., 2025)) editing models. The results, shown in table 5, demonstrate that our baseline consistently outperforms the other models across all three metrics. Further-

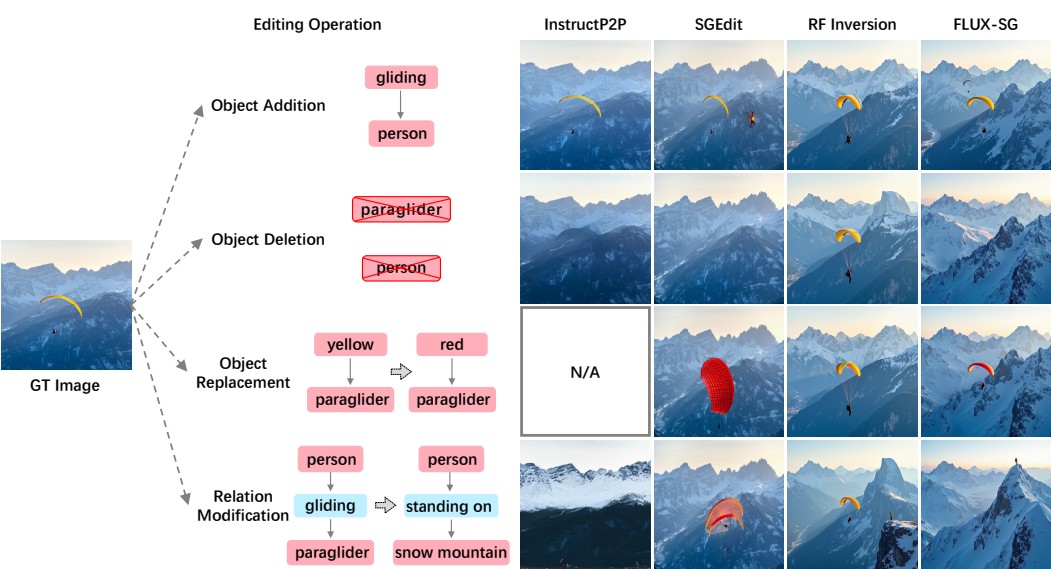

Figure 7: Case study of SG-based image editing.

more, we present case studies to visualize the effectiveness of SG-based image editing. Please refer to fig. 7 and Sec. A.1.3.

### A.1.3  QUALITATIVE RESULTS ON IMAGE EDITING

We conduct a case study to demonstrate the remarkable effectiveness of our model in image editing tasks, as illustrated in fig. 7. InstructP2P (Brooks et al., 2023) and SGEdit (Zhang et al., 2024c) perform correctly only on object deletion, but fail to adequately handle the other three editing types and occasionally cannot produce a valid output image. RF Inversion (Rout et al., 2025) exhibits strong image editing capabilities. However, it struggles with object-level edits, often producing incomplete object deletions and insufficiently controlled object replacements. We attribute these limitations to the loosely defined instructions in text-based editing baselines, which make it difficult to exert fine-grained control over the image content. In contrast, our FLUX-SG achieves precise and controllable results across all four types of image editing tasks, benefiting from the structured nature of the SG-based representation.

### A.2  EXPERIMENTAL SETUP

In this part, we introduce detailed setup in Sec. 5

**Implementation and Baselines.** We compare our baselines with SDXL (Podell et al., 2023), SG-Adapter (Shen et al., 2024), and SGdiff (Yang et al., 2022), following their evaluation settings. For SDXL-SG, we initialize the scene graph embeddings using OpenCLIP ViT-bigG/14 (Ilharco et al., 2021) and CLIP ViT-L/14 (Radford et al., 2021) in SDXL. The embeddings are refined with a 5-layer SG Encoder, each with 512 input and output dimensions. We augment FLUX-SG and SD3.5-SG following similar architectures but with 1024 hidden dimensions and initialization from their respective text encoders. In training, we employ Adam optimizer with a learning rate of 5e-4, training for one epoch on the full LAION-Comp dataset. SDXL-SG training is conducted on 8 NVIDIA RTX 4090D GPUs, while FLUX-SG and SD3.5-SG are trained on 4 NVIDIA A100 GPUs.

**Datasets and Evaluation Metrics.** We train existing models and our baselines on COCO-Stuff, Visual Genome (VG), and LAION-Comp datasets. We evaluate compositional image generation using FID for overall visual quality, CLIP score for similarity to the GT image, and SG-IoU, Entity-IoU, and Relation-IoU (Shen et al., 2024) to measure the consistency of generated scene graphs, objects, and relations against the real annotations, respectively. For SG-based image editing, we measure element composition (EC), relationship alignment (RA), and image quality (IQ), following SGEdit (Zhang et al., 2024c).

LAION Image          Original LAION Text          Scene Graph

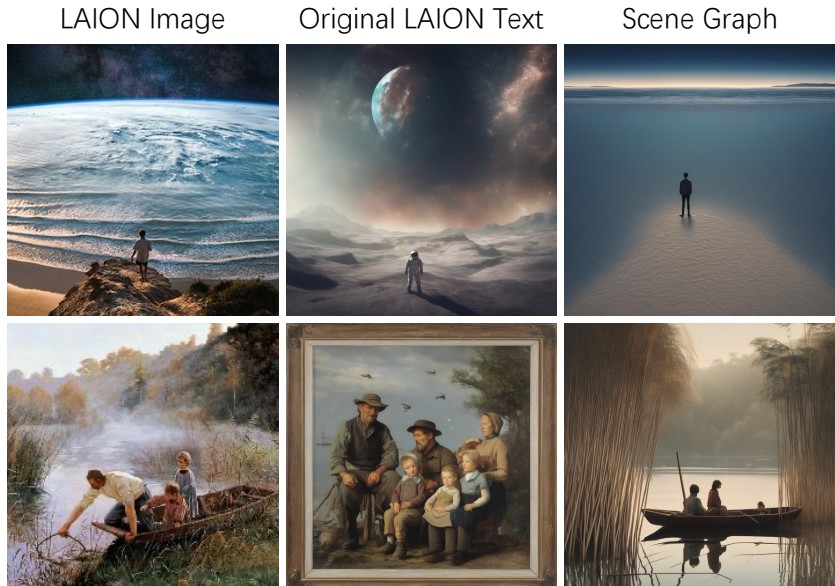

*Please select the image closest to the LAION image from those generated from the original LAION caption and the scene sraph.*

| Annotation | Original LAION Caption | Scene Graph |
|---|---|---|
| User Preference | 37% | 63% |

Figure 8: The result of user study. **Top:** We present images generated from original captions and scene graphs to users and ask them to choose the one that better aligns with the content of the LAION image. **Bottom:** Across 100 validation image pairs, users showed a strong preference for the results generated from scene graphs.

To evaluate the annotation quality, we propose SG-IoU+, Entity-IoU+, and Relation-IoU+ (Sec. A.7). Images are generated using scene graphs or LAION captions. The SG list, entity list, and relation list are then extracted by GPT-4o from both the generated and GT images. The consistency between the corresponding lists of the two images is calculated to assess the annotation accuracy. Due to the high cost, we compute the average for 300 samples as the result.

### A.3 USER STUDY

Beyond objective metrics, whether the results align with human cognition is also crucial. We conduct a user study to compare which annotation type generates images that better align with human perception.

We randomly select 100 text-sg-image triplets. In each trial, users are presented with three images: the LAION image and two images generated from the original LAION caption and the scene graph respectively. Users are asked to choose the image from the latter two that best matched the content of the LAION image. We invite 10 participants, with a 1:1 gender ratio and ages ranging from 20 to 30. They come from diverse backgrounds, including computer science, design, and human-computer interaction (HCI).

The result of user study is shown in fig. 8. A total of 63% of participants preferred the images generated from the scene graph, while only 37% chose those from the text prompt. This indicates that, compared to sequential text annotations, structured annotations have an overwhelming advantage in expressing image content.

**Notification to Human Subjects.** We present the notification to subjects to inform the collection and user of data before the experiments.

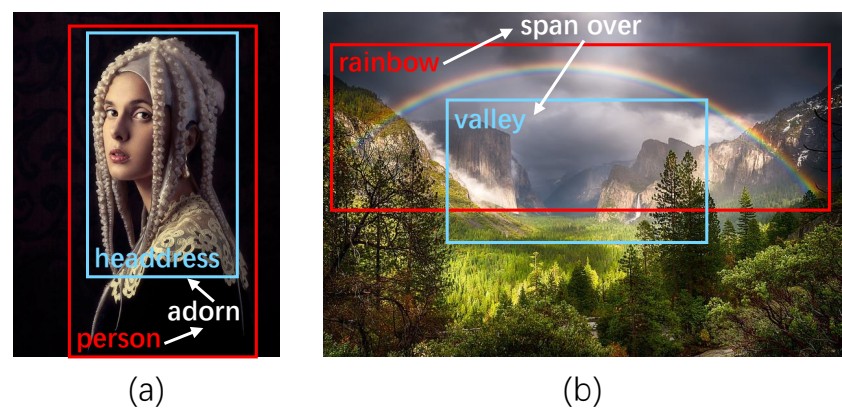

Figure 9: Example images from the LAION-Aesthetics dataset.

> Dear volunteers, we would like to express our thankfulness for your support to our study. We study an image generation algorithm, which translates scene graphs to realistic images.
>
> All information about your participation in the study will appear in the study record. All information will be processed and stored according to the local law and policy on privacy. Your name will not appear in the final report. When referred to your data provided, only an individual number assigned to you is mentioned.
>
> We respect your decision whether you want to be a volunteer for the study. If you decide to participate in the study, you can sign this informed consent form.

The use of users' data was approved by the Institutional Review Board of the main authors' affiliation.

## A.4 EXAMPLES OF LAION-COMP

Given an image, we employ a multimodal large language model, GPT-4o (OpenAI et al., 2024), to perform automated scene graph annotation. Our pipeline focuses on assigning distinct ids to different objects, identifying attributes for each object, labeling relations between objects, and adhering to other specified constraints. The annotations are strictly output in the designated format. Here, we provide two specific examples. For the image in fig. 9 (a), the highlighted portion corresponds to the following scene graph, with other parts omitted.

```
{
    "img_id": "482063",
    "name": "minus83166520...",
    "caption_ori": "Page 90 of Girl...",
    "score": "6.720815181732178",
    "url": "https://stories...",
    "items": [
        {
            "item_id": 0,
            "label": "person",
            "attributes": [
                "young",
                "female"
            ],
            "global_item_id": 3201686
        },
        {
            "item_id": 1,
            "label": "headdress",
```

```
                "attributes": [
                    "ornate",
                    "white"
                ],
                "global_item_id": 3201687
        },
        ...
    ],
    "relations": [
        {
                "triple_id": 0,
                "item1": 1,
                "relation": "adorn",
                "item2": 0,
                "global_relation_id": 2118510
        },
        ...
    ]
},
```

And for the image in fig. 9 (b), its highlighted portion corresponds to the following scene graph, with other parts omitted.

```
{
    "img_id": "483868",
    "name": "694108219422834467.jpg",
    "caption_ori": "Yosemite's Rainbow.  Yosemite National Park, California.",
    "score": "6.544332504272461",
    "url": "https://photos.smugmug.com/..."
    "items": [
        {
                "item_id": 0,
                "label": "rainbow",
                "attributes": [
                    "colorful",
                    "arc-shaped"
                ],
                "global_item_id": 3213781
        },
        ...
        {
                "item_id": 4,
                "label": "valley",
                "attributes": [
                    "green",
                    "vast"
                ],
                "global_item_id": 3213785
        },
        ...
    ],
    "relations": [
        {
                "triple_id": 0,
                "item1": 0,
                "relation": "span over",
                "item2": 4,
                "global_relation_id": 2126675
        },
```

| Complexity | Object Accuracy | Attribute Accuracy | Relation Accuracy |
|---|---|---|---|
| 0-10 | 98.5% | 96.1% | 95.0% |
| 10-20 | 99.7% | 97.6% | 95.7% |
| 20-30 | 99.0% | 98.4% | 95.6% |
| 30 and above | 98.1% | 98.0% | 96.6% |
| Average | 98.8% | 97.5% | 95.7% |

Table 6: The results of human verification. The complexity is define as the sum of the number of nodes and edges in a scene graph.

| Model | Complex | Spatial | Non-spatial |
|---|---|---|---|
| SDXL | 0.361 | 0.194 | 0.329 |
| SDXL-SG | 0.461 | 0.202 | 0.315 |

Table 7: Evaluation results of T2I model and our SG2IM baseline on T2I-CompBench.

```
        · · ·
    ]
},
```

### A.5 HUMAN VERIFICATION OF LAION-COMP

To investigate the accuracy of the automatically annotated dataset, we conduct a human verification. We randomly select 1,000 images from LAION-Comp and divide the samples into four categories based on the complexity of SG. The complexity is defined as the sum of the number of nodes and edges. A total of 20 users participate in the experiment, with a gender ratio of 1:1 and ages ranging from 20 to 30 years. The experiment requires users to examine the objects, attributes, and relations in each image and record the actual occurrences of them to compute the annotation accuracy. Detailed calculation is as follows:

$$\text{Accuracy} = \frac{\text{Actual Occurrences}}{\text{Occurrences in Annotations}} \tag{3}$$

This definition is similar to recall. As shown in table 6, the accuracies of object, attribute, and relation are 98.8%, 97.5%, and 95.7%, respectively, demonstrating the annotation of our dataset is accurate.

Generally, an annotation error rate of up to 5% (Fisher, 1970) is considered acceptable. According to the experimental results, the error of object, attribute, and relation are 1.16%, 2.5%, and 4.27%, respectively, all of which are well below the 5% error threshold. This suggests that the annotations in LAION-Comp are trustworthy and that the errors are within an acceptable range.

In conclusion, the human verification experiment validates the accuracy of our dataset, providing a reliable foundation for subsequent research.

### A.6 RESULTS ON T2I-COMPBENCH

In addition to the proposed CompSGen Bench, we also conduct experiments on T2I-CompBench (Huang et al., 2023). Since T2I-CompBench takes text as input, we first utilize GPT-4o (OpenAI et al., 2024) to convert the textual descriptions into scene graphs, which are then fed into SDXL-SG to generate images. As shown in table 7, our baseline outperforms the T2I model in both complex and spatial metrics, demonstrating the superior capability of SDXL-SG in handling complex scenarios and spatial relationships. The relatively lower performance on the non-spatial metric is due to the fact that in T2I-CompBench, the non-spatial score is directly determined by the CLIP score. Since the CLIP model is pretrained with extensive historical and abstract information, it incorporates additional contextual knowledge beyond the specific image content that the scene graph primarily focuses on.

Original LAION Caption     Scene Graph     Labeled Image

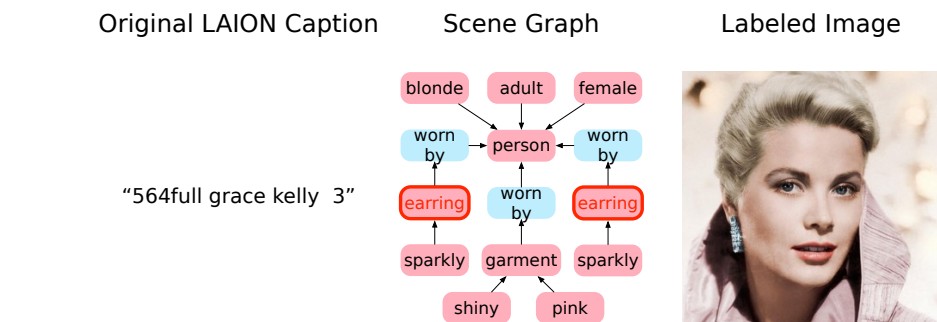

Figure 10: GPT-4o occasionally exhibits hallucination phenomena, labeling objects that do not exist in the image. For example, in a case where the image shows only one earring, GPT-4o incorrectly labels a nonexistent second earring. However, despite these issues, the overall quality of our annotations still surpasses that of LAION's original annotations.

## A.7 DETAILS OF ACCURACY METRICS

We leverage SG-IoU, Entity-IoU, and Relation-IoU (Shen et al., 2024) to measure the model's ability to generate complex scenes. Specifically, we use GPT-4 to extract scene graph lists from the generated images, with each list consisting of triples in the form $\langle s_n, r_n, o_n \rangle$. From this SG list, we derive the Entity and Relation lists and calculate the intersection over union (IoU) between the derived lists and the real annotations. Higher scores indicate stronger model capability in generating complex scenes.

Furthermore, we propose SG-IoU+, Entity-IoU+, and Relation-IoU+ to evaluate the annotation accuracy. Detailedly, we first generate two images: one using the original LAION captions and the other using scene graph from LAION-Comp. Then for the real image and the two generated images, we extract the lists of SGs, relations and entities from each image with GPT-4o again. Taking the lists of SGs as an example, the IoU scores is calculated between the list SG generated image and that of the real image. Also the IoU bewen the caption generated and the real is calculated. This IoU evaluates the extent that the generated images and the real image are similar along the SG structure, thus reflecting the annotation accuracy. It is the hight the better. Such IoU is also calculated on the lists of relations and entities.

## A.8 DISCUSSION ON ANNOTATION

### A.8.1 HALLUCINATIONS OF GPT-4O

In our annotation process, GPT-4o occasionally exhibits hallucination phenomena, generating information that does not actually exist. Through a random check of 100 annotation samples, we find that approximately 1% contain such issues. These issues typically manifest as annotations that do not strictly adhere to the image content but instead rely on semantic inference to incorrectly label objects that are not present. For example, in fig. 10, the GT image only shows one visible earring, while the other earring is occluded. However, GPT-4o erroneously infer its presence based on semantic reasoning.

Although the limitations of current multimodal large models make it challenging to completely avoid such problems, the quality of the original LAION annotation of the GT image in Fig. 10 is relatively low, further hindering the generation of complex scenes. Nevertheless, our annotation process strives to ensure the accurate description of entities and relationships within images, thereby maintaining a high overall annotation quality.

### A.8.2 DISCUSSION ON ANNOTATION ERRORS

It is inevitable that the automated annotation by multimodal large language model introduces a certain degree of error. We perform human manual check to show that this error remains within an acceptable range. Specifically, as shown in appendix A.5, the error rates for objects, attributes,

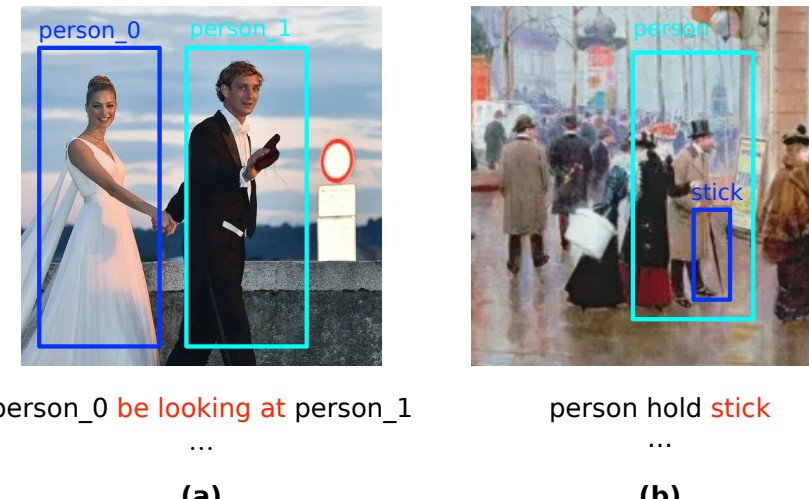

person_0 be looking at person_1
...

**(a)**

person hold stick
...

**(b)**

Figure 11: GPT-4o occasionally makes errors during the annotation process. For example, in (a), GPT-4o misidentifies the relationship, incorrectly assuming that person_0 is looking at person_1, which in fact is wrong. In (b), GPT-4o misclassifies a small object, labeling the item held by the person as a stick, when it is actually an umbrella.

and relations are 1.16%, 2.5%, and 4.27%, respectively. The error rate of some well-known datasets (Northcutt et al., 2021) exceeds 5%, while our error remains within an acceptable range. Furthermore, appendix A.8.1 indicates that approximately 1% of the samples exhibit annotation hallucination issues, where the multimodal large language model incorrectly infers labels that do not exist in the original image. Additionally, in appendix A.7, we introduce automated evaluation metrics for annotations, confirming that our scene-graph-based annotations more faithfully reflect image content compared to the original text-based annotations in the LAION dataset. All these experimental results substantiate the high reliability of our dataset annotations. Compared to the costly and time-consuming manual annotation process, the automated annotation approach proves to be efficient and feasible.

### A.8.3 FAILURE EXAMPLES

Through prompt engineering, we leverage GPT-4o to perform large-scale, high-quality scene graph annotations for images. While the majority of these annotations accurately describe the entities and their relationships in the images, a small number of errors still occur. In a random sample of 100 annotations, approximately 2% contain one mislabeled annotation, including inaccurate relationship descriptions (as shown in Fig. 11(a)) or entity recognition errors (as shown in Fig. 11(b)).

### A.8.4 ABSTRACT ATTRIBUTE

We follow the guidelines of previous work (Krishna et al., 2017) and try to ensure that attributes in the scene graph are abstract adjectives rather than specific objects.

### A.9 FURTHER DETAILS ON FOUNDATION MODELS

### A.9.1 A BRIEF INTRODUCTION OF GNN

In recent years, Graph Neural Networks (GNNs), a class of models originating from early work (Gori et al., 2005; Scarselli et al., 2008a), have emerged as a prominent area of research Duvenaud et al. (2015); Atwood & Towsley (2016); Bronstein et al. (2017); Monti et al. (2017). These networks offer a powerful and versatile methodology for learning representations from data structured as graphs. Consequently, GNNs are highly effective in domains where information can be modeled as a collection of nodes and edges, as they excel at capturing the intricate relationships between objects.

The applicability of GNNs is broad, with successful implementations in molecular chemistry for structure prediction (Gilmer et al., 2017), sociology for modeling social interactions (Welling & Kipf, 2016), and e-commerce for recommendation systems (Cen et al., 2020). Within computer vision, GNNs have been widely employed to encode the relational information inherent in scene graphs (Li et al., 2015; Hamilton et al., 2017). In this paper, we leverage the inherent structural processing capabilities of Graph Neural Networks (GNNs) to encode our structural annotations, effectively capturing their rich semantic and relational characteristics.

### A.9.2 A BRIEF INTRODUCTION OF STABLE DIFFUSION XL

SDXL (Stable Diffusion XL) (Podell et al., 2023) is an advanced latent diffusion model (LDMs) (Rombach et al., 2022) primarily designed for generating high-resolution images based on text prompts. It builds on the fundamentals of diffusion models (Ho et al., 2020) by utilizing a two-stage process: initially generating images from noise and then refining these images to enhance quality.

SDXL operates in a compressed latent space rather than the pixel space directly, using an autoencoder to encode an input image into a lower-dimensional latent space and then applying the diffusion process in this space. This approach is computationally efficient and enables the generation of high-quality, detailed images with fewer resources compared to pixel-based diffusion models (Ho et al., 2020). The model's architecture consists of an autoencoder and a UNet-based diffusion network that performs the denoising operations.

To interpret text prompts with high fidelity, SDXL integrates two text encoders (OpenCLIP ViT-bigG (Ilharco et al., 2021) and CLIP ViT-L (Radford et al., 2021)). These encoders convert the textual input into feature representations, which are then concatenated and used to condition the diffusion process, thereby allowing the model to follow text prompts more accurately.

The SDXL diffusion process is a series of denoising steps in which the model progressively reduces noise from an initial noise-filled image until a clear image is produced. This iterative process can be represented mathematically by

$$x_t = \sqrt{\alpha_t} \cdot x_0 + \sqrt{1 - \alpha_t} \cdot \epsilon \qquad (4)$$

Here $x_t$ is the noisy image at step $t$. $\alpha_t$ is a noise decay factor for each time step. $x_0$ represents the clean, noise-free image. And $\epsilon$ is random Gaussian noise added to the image at each step. Each step is controlled by a learned model, $\epsilon_\theta$, that predicts and subtracts noise from the image, allowing it to converge on a high-quality result as $t \to 0$.

SDXL's training objective is to minimize the mean squared error between the predicted noise and the actual noise added to the image. The conditional term is introduced through classifier-free guidance (Ho & Salimans, 2021), a mechanism that combines conditional information with the noise predictions from unconditional generation. This enables the model to better follow prompt details when generating images.

Specifically, the conditional loss function in SDXL can be represented as

$$\mathcal{L} = \mathbb{E}_{\mathcal{E}(x_0),c,\epsilon \sim N(0,I),t}[\| \epsilon - \epsilon_\theta(z_t, t, \tau(c)) \|_2^2], \qquad (5)$$

where $\mathcal{E}(x_0)$ and $z_t$ is latent representations of the original image and its noisy version at timestep $t$, $c$ and $\tau(c)$ is the input condition and its latent embedding, and $\epsilon_\theta(z_t, t, \tau(c))$ represents the model's noise prediction under condition $c$. Additionally, the conditional term $c$ includes spatial conditions like size and crop settings, enabling the model to adapt to various resolutions and framing needs. By minimizing this error, SDXL learns how to progressively remove noise and refine images accurately across various levels of initial noise.

To enhance the visual quality of generated images, SDXL includes a refinement model that operates in the latent space. This model further refines the output using SDEdit (Meng et al., 2022), an image-to-image process where noise is temporarily reintroduced and then denoised to improve quality.

### A.9.3 DETAILS OF SDXL-SG

Text-to-image (T2I) generation can produce highly detailed results. However, when the given text describes a relatively complex scene (e.g., an image containing multiple objects or multiple rela-

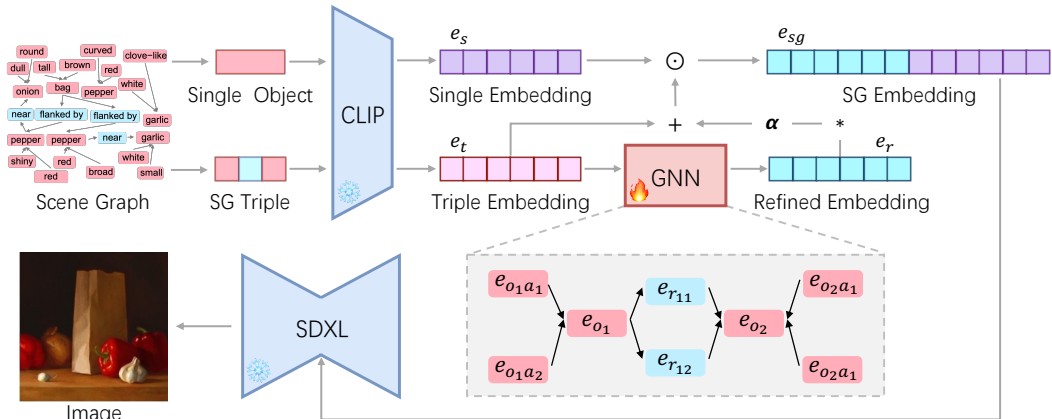

Figure 12: The architecture of our foundation model. Concatenation is indicated by $\odot$ and multiplication by $*$.

tionships between objects), the output of T2I models often falls short of expectations. Through experiments, we found that incorporating scene graph (SG) information during the image generation can significantly improve the model's ability to generate compositional images. Based on this observation, we introduce a foundation model to alleviate restrictions on text-to-image generation.

Our model is based on the SDXL (Podell et al., 2023) architecture, integrating SG information into the generation process through graph neural networks (GNN) (Scarselli et al., 2008b). As shown in fig. 12, we initialize each single object and triple of SG separately using the CLIP text encoder $E_T(\cdot)$ to get their embeddings $e_s$ and $e_t$. Specifically, for the triple embedding, it includes representations of objects $\mathbf{e}_{o_k}$, relations $\mathbf{e}_{r_{ij}}$, and object attributes $\mathbf{e}_{o_n a_m}$. Here, $\mathbf{e}_{o_k}$ represents the embedding of the $k$-th object in the SG, $\mathbf{e}_{r_{ij}}$ represents the embedding of the $j$-th word in the $i$-th relation, as some relations in LAION-Comp annotations may contain multiple words (e.g., "grown by"), and $\mathbf{e}_{o_n a_m}$ denotes the embedding of the $m$-th attribute word of the $n$-th object, as an object's attributes may consist of multiple words (e.g., "tall wooden building").

This structured SG input is then fed into the GNN. Objects serve as nodes, and relations act as edges. After that, we obtain a refined triple embedding $\mathbf{e}_r$, which can be represented as:

$$\mathbf{e}_r = \text{GNN}(E_T(triple^{sg})) \tag{6}$$

We introduce an $\alpha$ factor to control the strength of the refined triple embedding, ensuring stable learning for the model. The optimized triple embedding is represented as:

$$\mathbf{e}_{t'} = \mathbf{e}_t + \alpha \mathbf{e}_r \tag{7}$$

Finally, all triple embeddings are concatenated with single-object embeddings to form the SG embedding $\mathbf{e}_{sg}$, which is fed into the U-Net of SDXL for iterative noise prediction.

$$\mathbf{e}_{sg} = f(sg) = \text{concat}(\mathbf{e}_{t'}, \mathbf{e}_s) \tag{8}$$

We employ SDXL (Podell et al., 2023) as the pretrained framework. The model learns SG knowledge at time step $t$ by:

$$\mathcal{L} = \mathbb{E}_{\mathcal{E}(x), sg, \epsilon, t}[\| \epsilon - \epsilon_\theta(z_t, t, f(sg)) \|_2^2] \tag{9}$$

As introduction in appendix A.9.2, our training is conducted in the latent space to enhance efficiency. $f(sg)$ encapsulates the SG embedding output from SG encoder of our baseline. The training process dynamically adjusts parameters of SG encoder to minimize the gap between the predicted and added noise, which can reduce $\mathcal{L}$, improving the model's capability to handle compositional image generation.

Our architecture is designed to be lightweight and efficient. The generation time for 100 images at a resolution of $1024 \times 1024$ is measured. Our baseline model takes an average of 17.19 seconds per image, while the original SDXL model takes 16.70 seconds, both running on a single RTX 4090D GPU. Moreover, our SG encoder model has a parameter count of 14.70M, which is only 0.23% of

the approximately 6.6B parameters of the original SDXL, demonstrating its exceptional lightweight advantage. The inference time increases by less than 3%, and the parameter growth is negligible, making the additional computational cost almost insignificant. However, the improvement in output accuracy is substantial.

### A.9.4 DETAILS OF FLOW-MATCHING-BASED MODELS

Beyond SDXL, we further adapt our SG encoder to flow-matching-based generative backbones, specifically SD3.5-Medium (Stability-AI, 2024) and FLUX.1 Dev (Labs, 2024). Both architectures follow the paradigm of conditional flow matching (CFM), where the model learns a time-dependent vector field to transport noise samples toward the data distribution.

**Flow matching principle.** Given a clean image latent $x_0$ and Gaussian noise $\epsilon$, the rectified flow trajectory (Lipman et al., 2023; Liu et al., 2023; Albergo & Vanden-Eijnden, 2023) is defined as

$$z_t = (1 - t)x_0 + t\epsilon, \tag{10}$$

where $t \in [0, 1]$. The associated ground-truth vector field is

$$u_t(z|\epsilon) = \epsilon - x_0. \tag{11}$$

A neural network $v_\theta(z_t, t, f(sg))$ is trained to approximate $u_t$. The flow matching loss is given by

$$\mathcal{L}_{\text{FM}} = \mathbb{E}_{x_0,\epsilon,t} \left[ \| v_\theta(z_t, t, f(sg)) - (\epsilon - x_0) \|_2^2 \right]. \tag{12}$$

**Scene graph conditioning.** We retain the same GNN-based SG encoder introduced in appendix A.9.3, which transforms objects, relations, and attributes into SG embeddings. Specifically, single-object embeddings $\mathbf{e}_s$ and triple embeddings $\mathbf{e}_t$ are initialized using a pretrained CLIP encoder $E_T(\cdot)$. These are further refined by a GNN to obtain contextualized relation embeddings $\mathbf{e}_r$. As in SDXL-SG, we apply a learnable scaling parameter $\alpha$ to stabilize training, and concatenate the refined triples with object embeddings to form the SG representation:

$$\mathbf{e}_{sg} = f(sg) = \text{concat}(\mathbf{e}_t + \alpha\mathbf{e}_r, \mathbf{e}_s). \tag{13}$$

**Backbone-specific integration.** For SD3.5-SG, we adopt a two-level conditioning scheme. The pooled CLIP representation acts as a vector conditioning, while the unpooled token-wise embeddings are injected into the MM-DiT blocks for fine-grained alignment (Esser et al., 2024). For FLUX-SG, we inject SG embedding as the conditioning signal into the FLUX model's Denoising Diffusion Transformer (DiT) backbone. This injection is performed primarily through cross-attention and adaptive layer normalization (AdaLN) mechanisms within the transformer blocks.

**Training objectives.** The final training objective is flow matching with SG conditioning:

$$\mathcal{L} = \mathbb{E}_{x_0,\epsilon,t,sg} \left[ \| v_\theta(z_t, t, f(sg)) - (\epsilon - x_0) \|_2^2 \right], \tag{14}$$

where $f(sg)$ denotes the SG embedding. This objective is shared across SD3.5-SG and FLUX-SG, while the integration strategy differs as described above. By aligning SG embeddings with flow matching backbones, our model effectively enhances compositional generation capability.

### A.9.5 FLEXIBLE INPUT MODALITIES

To enhance user-friendliness, our framework is designed to support flexible, dual-modality inputs (free-form text or structured SGs) for compositional image generation. To this end, we introduce an automated text-to-scene-graph pipeline underpinned by Qwen3 (Yang et al., 2025), capable of precisely converting even free-form text into structured SGs, as illustrated in fig. 17. We have rigorously validated the accuracy and reliability of this text-to-SG conversion. A comprehensive description of the pipeline and the validation experiments is provided in Sec. A.10.

### A.9.6 ADDITIONAL RESULTS

**Successful and Failure Examples.** We provide additional experimental results. fig. 13 presents more successful cases, while fig. 14 shows some failure cases, including object misalignment, incorrect object shape generation, and errors in object appearance generation.

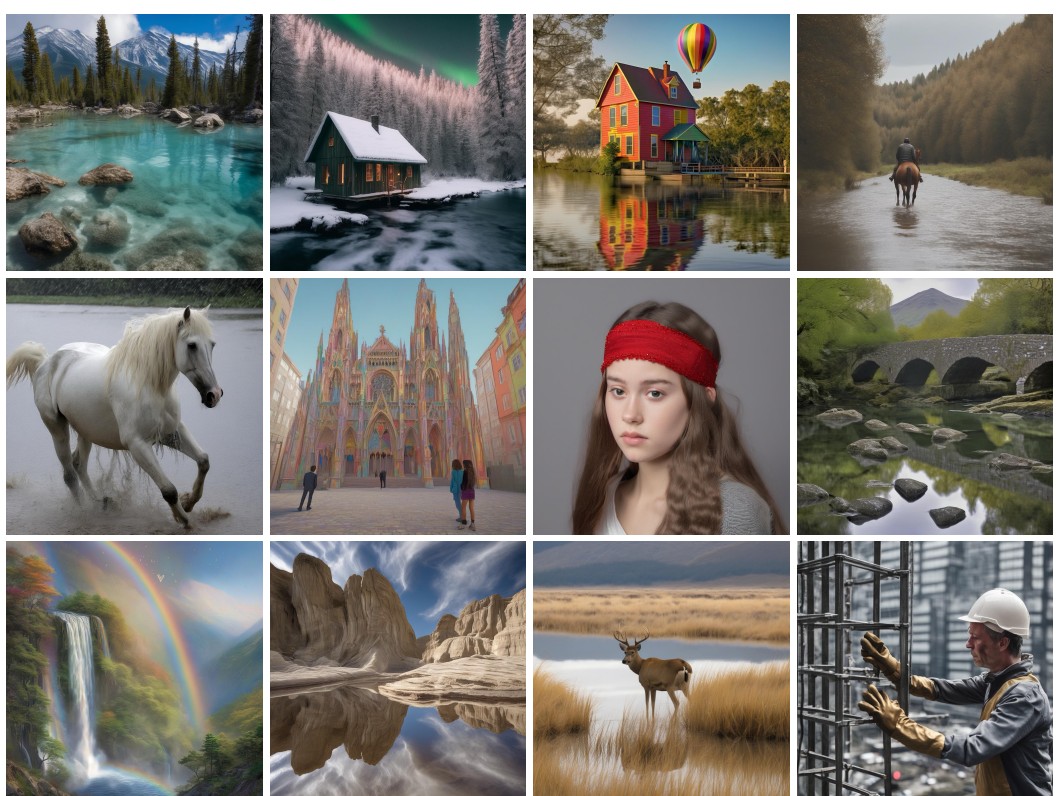

Figure 13: Additional results of successful examples.

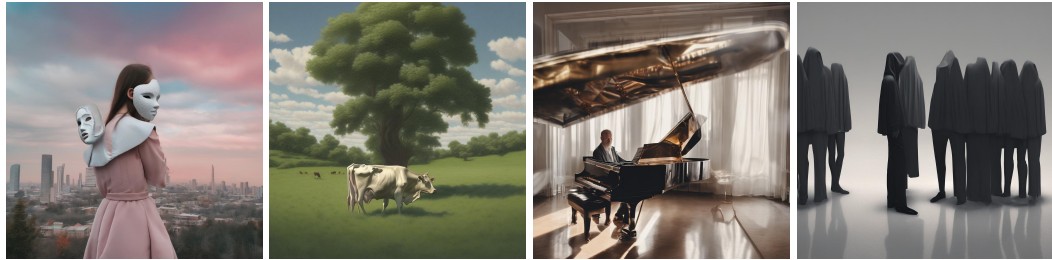

Figure 14: Additional results of failure examples.

**Examples of different complexities.** We define the complexity of a scene graph as the sum of the number of nodes and edges. This definition is consistent with text length as an intuitive measure of prompts. For scene graph inputs of varying complexities, our baseline model can accurately recognize and faithfully generate the corresponding outputs. As shown in fig. 15, for two scenes with a complexity of 3 and one scene with a complexity of 6, despite the content being similar but with different complexities, SDXL-SG consistently achieves accurate and high-quality faithful generation.

**Visualization of Sensitivity to Similar Content.** Our model is sensitive to subtle semantic differences. As shown in fig. 16, although the input scene graphs are highly similar, resulting in embeddings with a high degree of similarity, our baseline model can still accurately distinguish between them and correctly generate the corresponding content.

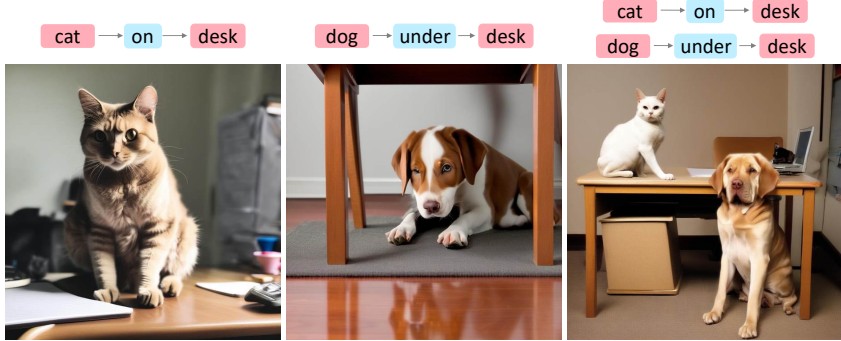

Figure 15: Additional results regarding different complexities with similar content.

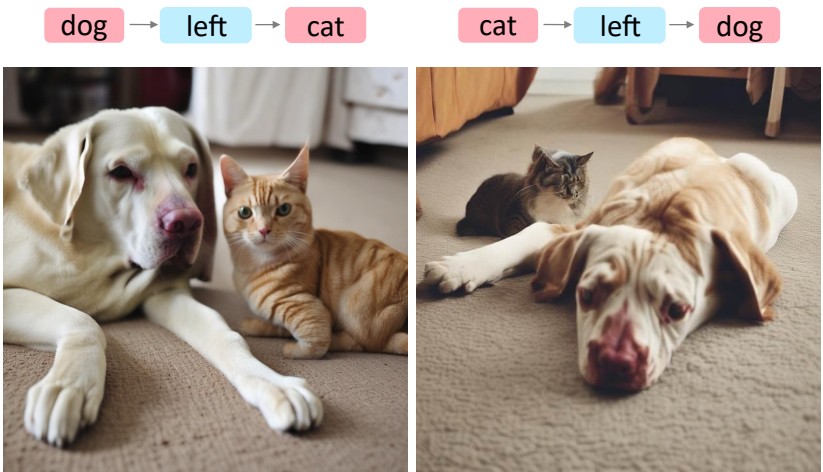

Figure 16: Additional results regarding sensitivity to similar yet semantically different content.

### A.10 TEXT-TO-SCENE-GRAPH PIPELINE

Figure 17 shows our text-to-scene-graph pipeline. It adheres to a strict zero-inference policy, ensuring that only explicitly stated information is extracted, with high fidelity to the original input and standardized JSON-compliant output. During object extraction, each mentioned entity is assigned a unique identifier following an incremental numbering convention, with duplicate categories receiving distinct IDs, and all objects—including orphan nodes—retained for completeness. Attribute mapping is confined to explicitly stated adjectives or states, preserves their original order of appearance, and formats them into sub-arrays associated with corresponding object IDs. Relation extraction identifies explicit action and spatial relations based on verb-preposition cues, represents each as a triplet structure $[subject\_id, relation, object\_id]$, and applies deduplication to retain only the first occurrence of redundant relations. The pipeline further enforces restriction policies that prohibit inferred relations, abstract concepts, synonym substitution, and compound noun splitting. Validation mechanisms verify ID continuity, ensure all attribute and relation references are valid, and perform deduplication checks across triplet fields. The result is a structured and semantically consistent scene graph that can be directly utilized by downstream generative models, effectively bridging natural language expressivity and structured semantic representation.

Furthermore, we rigorously validate the accuracy of the proposed pipeline. Specifically, we first employ VLM-based prompt engineering to generate a detailed textual description of an image, denoted as $txt_1$. This description is then converted into a structured scene graph representation using the aforementioned text-to-SG conversion pipeline. To evaluate the alignment between the generated scene graphs and the original images, we conduct a user study involving 20 participants and

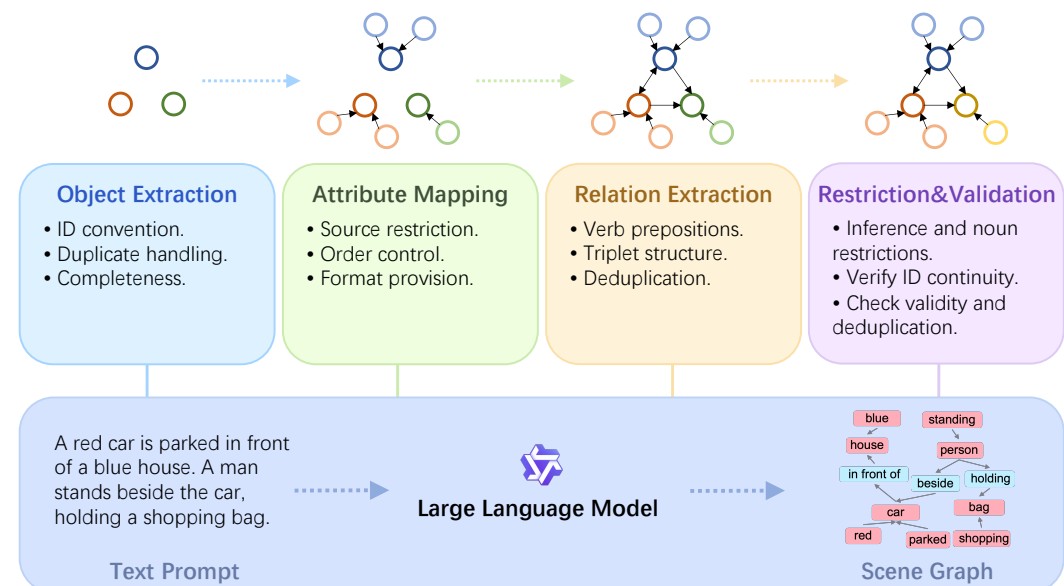

Figure 17: A pipeline for converting text prompts into scene graphs.

1000 images. The experimental results demonstrate high performance, with accuracies for object, attribute, and relation extraction reaching 95.3%, 92.2%, and 92.9%, respectively.

However, since the overall image-to-SG process consists of two stages—VLM-based image captioning followed by our text-to-SG conversion—we perform an additional ablation to ensure that the high accuracy is indeed attributable to our pipeline. Specifically, we convert the resulting scene graph back into a textual description, denoted as $txt_2$, using a analogous LLM-based inversion pipeline. We then compute the CLIP similarity between $txt_1$ and $txt_2$ across a set of 1000 samples, yielding an average CLIP score of 0.843. This high similarity score confirms the reliability and accuracy of our text-to-SG conversion pipeline. The minor errors observed are primarily due to information loss during the initial VLM-based image-to-text conversion step, rather than inaccuracies within our pipeline. This result underscores the effectiveness of our method in robustly translating user-provided textual prompts into structured and semantically faithful scene graph representations.

## A.11 DISCUSSION ON COMPLEX SCENE GENERATION

Complex scene generation is a challenging task attracting attention from the community of image generation. Compositional Diffusion (Liu et al., 2022) breaks down complex text prompts into multiple easily generated segments, but it is limited to conjunction and negation operators. Attend-and-Excite (Chefer et al., 2023) guides pre-trained diffusion models to generate all entities in the text through immediate reinforcement activation, yet it still faces attribute leakage issues. MIGC (Multi-Instance Generation Controller) (Zhou et al., 2024b) and MIGC++ (Zhou et al., 2024a) adopts a strategy of generating individual instances separately and then integrating them, while incorporating multimodal descriptions for attributes (text and images) and localization (bounding boxes and masks). By incorporating appearance tokens and an instance semantic map, IFAdapter (Wu et al., 2024c) enhances the fidelity of fine-grained features in multi-instance generation while ensuring spatial precision. 3DIS (Zhou et al., 2024c) decouples the multi-instance generation task into two stages: depth map generation and detail rendering. By combining depth-driven layout control with training-free fine-grained attribute rendering, it significantly enhances instance positioning accuracy and detail representation.

These works effectively address the challenges of multi-instance compositional generation. However, they primarily control the generated objects at the spatial level and fail to resolve inaccuracies in generating abstract semantic relationships between objects, such as "holding" or "riding".

In contrast, Wang et al. (2024b) disentangles layouts and semantics from scene graphs, leveraging variational autoencoders and diffusion models to significantly enhance instance relationship mod-

eling and fine-grained control in complex scene generation. This approach enhances the model's understanding and representation of abstract semantics through the structured form of scene graphs. Nevertheless, it still faces generation bottlenecks due to dataset limitations.

Therefore, we propose the LAION-Comp dataset to fundamentally address the challenges of complex scene generation at the data level. Simultaneously, we introduce a baseline model that enables simple and efficient generation of complex scenes based on scene graphs.

### A.12 ADDITIONAL EXPLANATIONS

#### A.12.1 INFERENCE FOR DIFFERENT INPUT MODALITIES

In our experimental section, we provide a comprehensive comparison between T2I models and the SG2IM models. Here, we explain the inference process for different input modalities. For the T2I model on LAION-Comp, we semantically concatenate the scene graph into text and use the T2I model to generate images. For the SG2IM model, the scene graph embedding is added to the original CLIP embedding (appendix A.9.3) as the generation condition. The CLIP Score in the experimental table measures the similarity between the generated image and the ground truth image. IoUs are calculated based on the scene graph derived from the same image labels, ensuring that all comparisons are made under fair conditions.

#### A.12.2 ACQUISITION OF SCENE GRAPHS

Currently, there are many studies on Scene Graph Generation (SGG), which provides an effective approach for obtaining scene graphs. Additionally, we can leverage multimodal large language models to generate corresponding scene graphs based on given content. Furthermore, there are existing interactive scene graph annotation visualization tools (Ashual & Wolf, 2019), which are highly convenient for editing, such as adding, removing, or modifying scene graph elements. Compared to text-based input formats, scene graphs are more structured, easier to edit, and simpler to construct.

#### A.12.3 THE REASON TO ANNOTATE LAION-AESTHETIC

For several reasons, we choose LAION-Aesthetic as the foundation for constructing a complex scene graph dataset. First, LAION-Aesthetic offers high visual quality, which is important in image generation. Datasets initially intended for detection or segmentation are mostly obtained from ordinary photographs and may not exhibit high visual quality. Second, the dataset contains a rich variety of scenes, which is crucial for compositional image generation. Finally, compared to other datasets, LAION-Aesthetic has a high aesthetic score, representing a higher data benchmark.

#### A.12.4 DISCUSSION ON DATA VALIDITY

We argue the improvement in model performance is attributed to the quality of the data and additional training, rather than a mere increase in data volume. The data in the LAION dataset has already been adopted in the training of SDXL, so the performance enhancement is not a result of the larger dataset size. The model's improved performance is from the enhanced accuracy and comprehensiveness of the annotations, which reflects the advanced quality of the dataset.

### A.13 LIMITATION

We summarize statistics on the types of objects annotated in the LAION-Aesthetics (Schuhmann et al., 2022) and LAION-Comp datasets. Among 10,000 samples, LAION-Aesthetics contains 12,263 distinct object types, which reduces to 5,811 after excluding proper nouns. In comparison, LAION-Comp includes 1,429 types, all of which are common words without any proper nouns. This difference reflects a limitation of LAION-Comp, as its vocabulary distribution is relatively less extensive. Furthermore, since LAION-Comp focuses on scene graph that describe specific content within images, it is less sensitive to abstract cues such as historical context or stylistic elements. Integrating these control factors into the scene graph-to-image process remains a promising direction for future research.

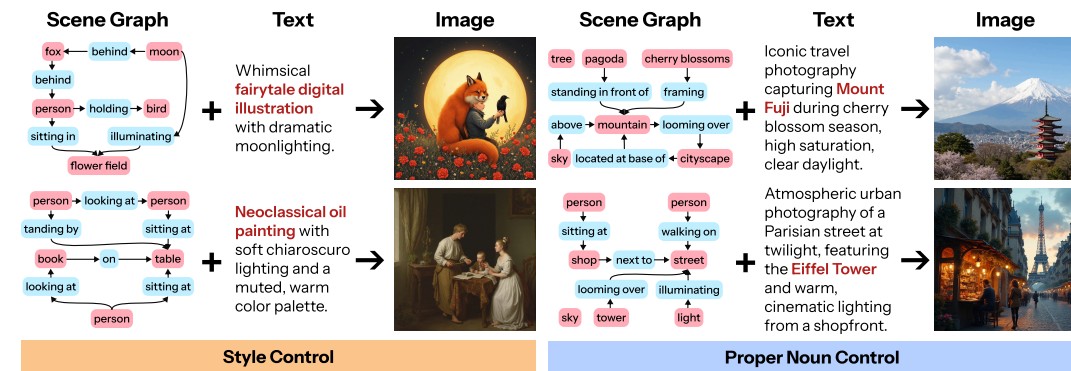

Figure 18: Extending proper noun and style control via scene graph–text integration.

Compared to manual annotation workflows, automated method significantly improves efficiency and reduces costs. However, it slightly lacks the precision achievable through human annotation, as discussed in appendix A.8.1 and appendix A.8.3. This limitation, however, is an inherent aspect of automated processes.

When annotating images with scene graphs, we employed a multimodal large language model rather than existing scene graph generation (SGG) models. One reason for this choice is the bounding box constraint—training SGG models typically requires bounding box information, which are not available in LAION-Comp. However, as demonstrated in appendix A.5, appendix A.8.1, and appendix A.7, our designed automated annotation approach can accurately generate scene graph (SG) annotations, ensuring that this limitation does not significantly impact the contributions of our work. Nevertheless, we will consider incorporating bounding box and instance segmentation enhancement in future research.

### A.14 PROPER NOUN AND STYLE CONTROL

We investigate a hybrid conditioning strategy that integrates our structural Scene Graph (SG) embeddings with the embeddings of descriptive text. This approach extends SG-based image generation, which typically focuses on realistic content, to a broader scope of proper noun and style control. As illustrated in fig. 18, it allows the model to faithfully adhere to the content interactions defined by the SG while leveraging text prompts to inject long-tail concepts and stylistic attributes that refine the generated output.

Specifically, we demonstrate Proper Noun Control by pairing a scene graph with specific location/building proper nouns, such as "Mount Fuji" or "the Eiffel Tower". The results show that the model preserves the rigorous structural relationships while accurately rendering the specific architectural features of the requested landmark, rather than a generic building. Similarly, for Style Control, the generated images maintain the semantic layout and interaction logic consistent with the SG, while simultaneously injecting the corresponding color palettes, lighting, and textures derived from the textual description.

These findings confirm that our structural conditioning is complementary to text-based generation, offering a flexible interface where users can enforce rigid structural constraints via graphs while recovering enhanced expressiveness and stylistic diversity through text prompts.

### A.15 GENERALIZATION ANALYSIS

We conduct a multi-faceted generalization analysis to mitigate the issue of training and testing on the same data distribution. First, the test set used to evaluate our compositional generation metrics—namely, SG-IoU, Entity-IoU, and Relation-IoU—is a mixture of different distributions. A total of 300 samples were procured by randomly selecting 100 images each from COCO, VG, and LAION-Comp. This balanced composition ensures the fairness of the evaluation.

Second, table 2 presents the results of a baseline trained on COCO and VG but tested on the completely separate LAION-Comp test set. Furthermore, table 7 reports our results against existing

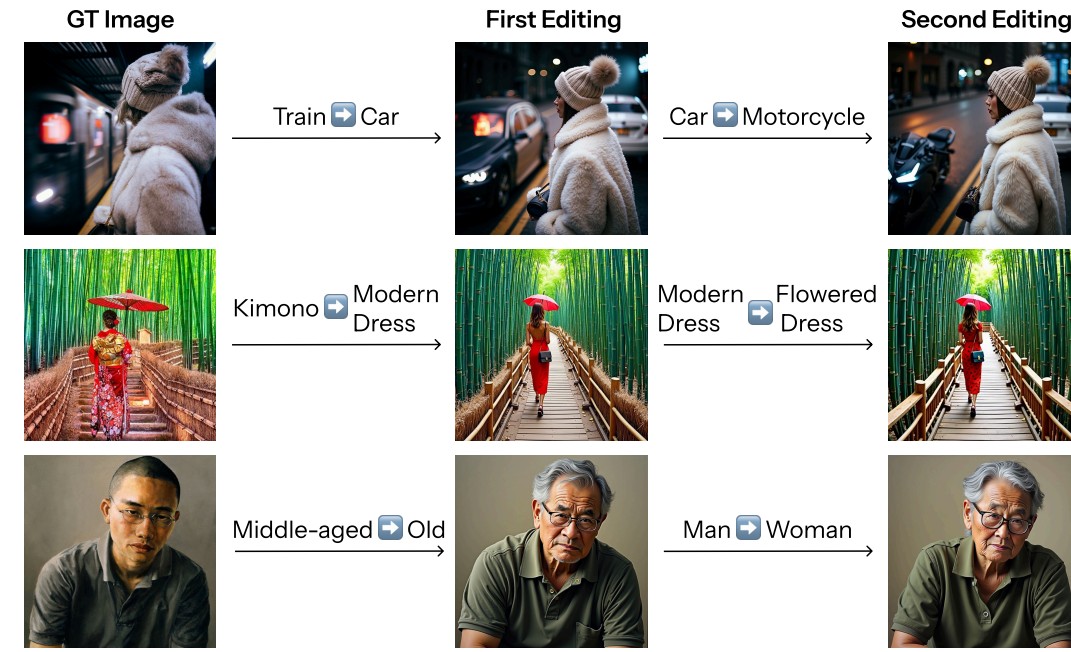

GT Image      First Editing      Second Editing

Figure 19: Visualization of multi-step editing results.

baselines on T2I-CompBench, an entirely independent, external benchmark. These evaluations collectively and effectively prevent the issue of the training and testing sets sharing the same distribution, thereby validating the reliability of our proposed data and method.

### A.16 EXPERIMENTS OF LOCAL AND MULTI-STEP EDITING

We conduct both qualitative and quantitative experiments to evaluate the editing locality and multi-step stability of our method.

**Qualitative Evaluation.** For each image, we apply two edits to the same object region. As illustrated in fig. 19, the visual differences remain largely confined to the edited area. Despite performing multiple edits, the unedited regions exhibit strong preservation. This demonstrates that our method achieves high editing locality, ensuring that changes remain constrained.

**Quantitative Evaluation.** We further perform a quantitative analysis on our 30-image editing test set. Each image undergoes two rounds of object replacement, and we compute the LPIPS score (Zhang et al., 2018) between the edited output and the ground-truth image, using SqueezeNet as the feature extractor. The LPIPS scores are 0.281 for the first edit and 0.312 for the second. These low scores indicate strong fidelity to the original image, confirming that the edits are locally restricted and that the results remain stable even after multiple editing steps.

### A.17 MULTI-OBJECT EDITING

Figure 20 illustrates the multi-object editing capability of our method. When simultaneously editing multiple objects, relations, and attributes in a single image, the image can be generated with high quality while adhering to complex multi-editing operations. This visually and explicitly demonstrates the robustness of both our data and approach.

### A.18 ANNOTATION BIAS ANALYSIS

We conduct a comprehensive annotation bias analysis to further explore the influence of different error modes on annotation accuracy. We validate 1,000 annotated samples and identified their respective annotation error types, with the results detailed in fig. 21.

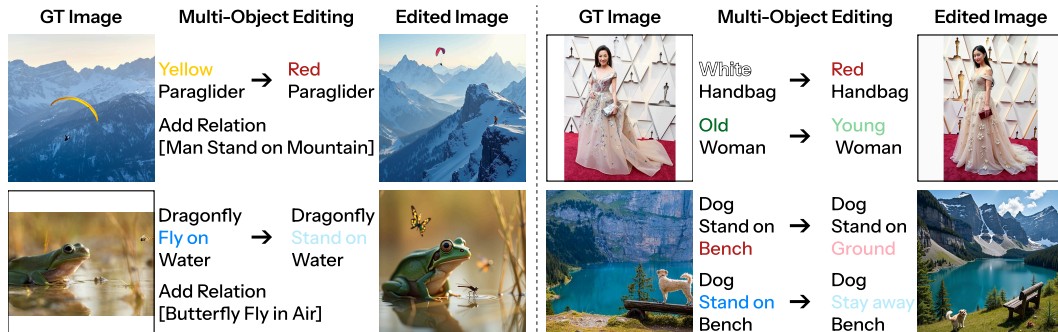

Figure 20: Case study of multi-object editing, involving multiple objects, relations, and attributes.

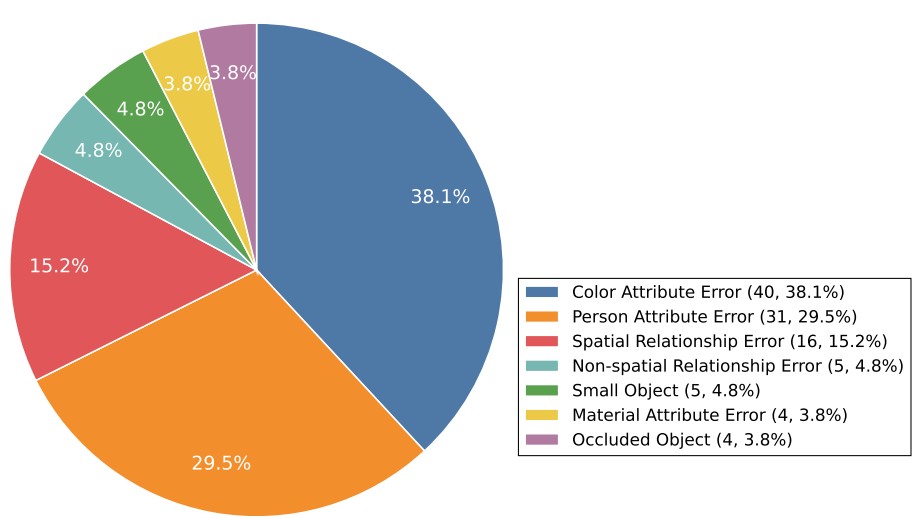

Figure 21: Distribution statistics of annotation error types.

The most frequently occurring error is the color attribute error, followed by the person attribute error and the spatial relationship error, which occupy the top three modes. These three categories collectively account for $82.8\%$ of the total observed errors. In addition, other factors contributing to annotation inaccuracy include the non-spatial relationship error, small object error, material attribute error, and occluded object error. Notably, even the most prevalent error, the color attribute error, only accounts for $4\%$ (40 out of 1,000 samples) of the total samples examined, thus confirming the high reliability of our annotations.

Considering the substantial labor and resource expenditure, we utilize a Vision-Language Model (VLM), specifically Qwen3-VL (Yang et al., 2025), to assist human annotators primarily with relationship identification. However, the resulting outputs are subjected to and confirmed by rigorous human verification.

## A.19 SOCIAL IMPACT

Scene graph to image generation holds great potential to benefit diverse fields, from content creation and education to virtual reality and simulation. By enabling the generation of realistic images from structured descriptions, this technology democratizes creative processes, allowing individuals with limited artistic skills to visualize complex ideas efficiently. Moreover, it can facilitate accessibility for users with disabilities, providing new ways to interact with visual content.

However, the technology also poses challenges, such as potential misuse for generating misleading, harmful content and negative bias. To mitigate these risks, the dataset and methods proposed in this work prioritize ethical considerations, including content moderation and bias reduction. Future research and collaboration across disciplines are essential to ensure that such technologies align with societal values while maximizing their positive impact.

## A.20 DETAILS ON LLM USAGE

During the preparation of this manuscript, we utilize a Large Language Model (LLM) to assist with language translation and refinement. The prompt template employed is: "Please translate the following into academically rigorous English."

