# OpenReview forum: "LAION-Comp: Unlocking Controllable and Compositional Generation with Structural Annotations"
_ICLR.cc/2026/Conference — Submitted to ICLR 2026_

### Official Review · Reviewer_iUAg · 2025-10-23

**Soundness:** 3
**Presentation:** 3
**Contribution:** 2
**Rating:** 4
**Confidence:** 4

**Summary:**

In this paper, the authors construct a large-scale scene-graph (SG) dataset (LAION-Comp) with annotations of high-aesthetics LAION images via a MLLM pipeline with partial human verification. And the authors use a GNN-based SG encoder to embed scene-graph annotation so that inject SG embeddings into diffusion and flow-matching backbones for controllable image generation. Compared with original text counterparts, the proposed richer scene-graph annotation leads to more precise annotation-following image generation.

**Strengths:**

1. Large-Scale Scene-graph Dataset. In this paper, the authors construct a dataset pipeline with MLLM (GPT-4) to produce a large-scale txt-scene-graph dataset for compositional generation, which can contribute to the incremental investigation of scene-graph-based methods in the community.

2. Good Performance. The authors demonstrate that the mainstream image generation backbones fine-tuned on the proposed dataset can achieve superior performance in complex scene synthesis.

**Weaknesses:**

1. Concerns about Overclaim. The claim of “our work represents a pioneering effort in annotating complexity on existing image datasets…” is not proper and misleading. It is obvious that  COCO Stuff and Visual Genome are earlier works on scene-graph annotation.

2. Limited Annotation Diversity Improvement. Except for larger scale and richer per-node/edge details, the dataset’s annotation scheme lacks variety: it sticks to the standard scene-graph paradigm seen before. Consequently, there is limited conceptual insight here—the advances are incremental in data scope rather than introducing a more expressive or innovative annotation form.

3. Underdeveloped Benchmark Justification. For the proposed CompSGen Bench, the paper clearly defines sample selection and reports FID/CLIP alongside SG-IoU/Entity-IoU/Relation-IoU, but it does not demonstrate why this benchmark evaluates better than existing ones—there is no systematic comparison on human correlation, robustness to parser/threshold choices, or justification of the chosen complexity threshold. This makes the benchmark feel like a reasonable task specification rather than a fully argued contribution with demonstrated measurement advantages.

4. Lack further exploration of the scene-graph relation among objects. Although the dataset is dominated by non-spatial relations, the paper reports only aggregate metrics. Without relation-type–stratified analyses or stress tests, it remains unclear whether spatial or non-spatial relations are better controlled, and which SG components actually drive controllability.

5. Lack of multiple Object Editing Results. With richer scene-graph annotations and data, robustness to multiple object editing about relation and attribute should be verified to further evaluate the enhanced large-scale dataset.

**Questions:**

Please refer to the weaknesses.

---

> ### Author Response · Authors · 2025-12-02
>
> To  Reviewer iUAg,
>
> You are a remarkably kind reviewer, and we truly appreciate your thoughtful and positive assessment of our work. We value your recognition of LAION-Comp as a "large-scale scene-graph dataset" and your assessment that our model achieves "superior performance" in complex scene synthesis. We thank you for your constructive criticism regarding the framing of our contribution and the depth of our evaluation. Below, we address your specific concerns.
>
> **[W1]  Clarification on "Pioneering" Claim.**
>
> **[Response to W1]**
>
> We apologize if the phrasing "pioneering effort" appeared to overlook foundational works such as COCO-Stuff and Visual Genome (VG). This was certainly not our intention. We cite and thoroughly discuss COCO and VG as the cornerstones of this field (Sec. 2, Table 2).
>
> By "pioneering," we specifically intended to highlight the effort to scale structurally complex annotations to high-quality, large-scale images (LAION), thereby bridging the gap between small, manually annotated structured datasets (VG/COCO) and large but unstructured text-only datasets (LAION-5B).
>
> We have revised the corresponding text to: "Our work represents a significant step toward scaling structurally complex annotations to high-quality, large-scale datasets, enabling broader scene synthesis and editing."
>
> **[W2]  Annotation diversity and paradigm.**
>
> **[Response to W2]**
>
> We appreciate the reviewer's comment regarding the annotation form. However, we respectfully disagree with the premise that adhering to a standard paradigm limits conceptual insight. We argue that data scale, density, and semantic distribution are transformative drivers of innovation, often more so than novel annotation schemas.
>
> 1. Our work can be compared to the "ImageNet Moment" for Scene Graphs. History in computer vision demonstrates that "incremental" improvements in data scope often trigger fundamental breakthroughs. Before ImageNet, datasets like Caltech-101 or Pascal VOC used standard classification labels and bounding boxes. ImageNet[1] did not invent a new annotation form; it utilized the existing WordNet hierarchy. However, by scaling up to 1M+ images and 1000 classes, it shifted the community's focus from hand-crafted features to deep representation learning. Similarly, LAION-Comp aims to be the "ImageNet" for compositional generation. Existing SG datasets (Visual Genome, COCO-Stuff) are limited in scale and density, causing T2I models to fail at complex logic. By scaling high-quality SG annotations to 540K+ aesthetic images, we provide the critical mass needed to unlock capabilities (e.g., handling 4+ objects with complex relations) that were previously impossible with smaller datasets.
>
> 2. The reviewer notes "richer per-node/edge details" as a minor point, but we argue this is the core conceptual advancement. Current T2I models suffer from semantic sparsity rather than new annotation form. We shift the distribution from spatial-heavy (VG: 58% spatial, Line 302.5) to interaction-heavy (LAION-Comp: 77% non-spatial actions like "playing", "wearing", Line 301-302). This is not just "more details"; it is a qualitative shift in the learning objective. It forces the model to learn abstract physical and functional interactions rather than simple spatial layout.
>
> [1] Deng, Jia, et al. Imagenet: A large-scale hierarchical image database. 2009 IEEE conference on computer vision and pattern recognition. Ieee, 2009.
>
> **[W3] Benchmark Justification.**
>
> **[Response to W3]**
>
> First, our benchmark is superior to existing ones in terms of data quality and scale. Prior benchmarks typically lack the high aesthetic quality of our images, and their structural datasets are substantially smaller than ours. We adopt widely used metrics, including FID and CLIP, alongside SG-IoU, Entity-IoU, and Relation-IoU. The combination of reliable metrics and higher-quality data demonstrates the superiority of our benchmark.
>
> Regarding the rationale behind the complexity threshold, please refer to Fig. 1. The threshold of >4 relations was chosen based on th e observation (Fig. 1) that standard T2I models (e.g., SDXL) undergo a notable performance drop once the relational complexity exceeds this point.
>
> In addition, we validated human correlation. As shown in Sec. A.3 (Line 1020-1022), our user study reports a 63% human preference for SG-generated images, which exhibits strong correlation with our automated metrics (e.g., SG-IoU). This confirms the relevance and validity of our benchmark.

---

> ### Author Response · Authors · 2025-12-02
>
> **[W4] Stratified Relation Analysis.**
>
> **[Response to W4]**
>
> We have performed a stratified analysis of both **annotation quality** and **generation performance**.
>
> For the dataset annotations, we conducted stratification across spatial vs. non-spatial relations, scene-graph components, and complexity levels. As shown in Sec. 3.2 (Line 301–303), LAION-Comp is dominated by non-spatial relations, whereas VG is conversely skewed toward spatial ones, as summarized in the table below. LAION-Comp therefore captures more abstract, functional, and interaction-centric semantics, moving beyond the primarily geometric or locational emphasis of VG.
>
> | Dataset        | Spatial Relation | Non-spatial Relation |
> |----------------|------------------|-----------------------|
> | Visual Genome  | 58.02%           | 41.98%                |
> | LAION-Comp     | 22.52%           | 77.48%                |
>
> As reported in Sec. A.5 (Line 1167-1167), and shown in Table 6, under human validation the object, attribute, and relation accuracies of LAION-Comp are 98.8%, 97.5%, and 95.7%, respectively. Moreover, these accuracies do not vary significantly across different complexity levels, demonstrating the reliability of our dataset.
>
> For generation performance, we additionally conducted isolated evaluations for spatial and non-spatial relations, with the results summarized in the table below. The absence of significant degradation across simple and complex relations confirms the effectiveness of both our data and our method in handling complex semantics.
>
> | Relation Type | SG-IoU | Entity-IoU | Relation-IoU |
> |---------------|--------|-------------|----------------|
> | spatial       | 0.357  | 0.857       | 0.752          |
> | non-spatial   | 0.467  | 0.835       | 0.843          |
>
> Furthermore, Table 7 reports results on T2I-CompBench. Our SDXL-SG substantially outperforms the SDXL baseline on the Complex metric (0.461 vs. 0.361) and also improves the Spatial metric (0.202 vs. 0.194). The larger gain in the Complex category, which involves attribute binding and non-spatial interactions, suggests that the explicit edge modeling in our GNN is the primary driver of controllability for abstract concepts.
>
> | Model  | Complex | Spatial |
> |---------|---------|---------|
> | SDXL    | 0.361   | 0.194   |
> | SDXL-SG | 0.461   | 0.202   |
>
> **[W5] Multi-Object Editing.**
>
> **[Response to W5]**
>
> Our graph-based interface naturally supports multi-object editing because the input condition is the embedding of the entire scene graph $e_{sg}$. Modifying multiple nodes (e.g., changing "cat" $\to$ "dog" and "table" $\to$ "chair") is handled as a single, coherent structural update. In Fig. 7, we demonstrate various types of editing. Although the figure presents isolated operations for clarity, the "Relation Modification" example (changing "gliding" to "standing on" + "snow mountain") inherently involves interactions among multiple entities.
>
> In addition, we performed a new multi-object editing qualitative experiment that explicitly demonstrates simultaneous multi-entity/relation/attribute modifications to verify robustness. Please refer to Sec. A.17 Multi-Object Editing (Line 1771-1776) for details.

---

### Official Review · Reviewer_VzmP · 2025-10-30

**Soundness:** 2
**Presentation:** 3
**Contribution:** 2
**Rating:** 4
**Confidence:** 3

**Summary:**

This paper introduces LAION-Comp, a large-scale dataset of 540K LAION-Aesthetics images annotated with detailed scene graphs (objects, attributes, relations) produced by GPT-4o and partially human-verified. The authors design a graph neural network (GNN) scene-graph encoder that conditions diffusion and flow-matching text-to-image backbones (SDXL, SD3.5, FLUX) to produce SG-conditioned variants (e.g., SDXL-SG, FLUX-SG). They also propose CompSGen Bench, a 20,838-sample test split focusing on complex scenes (>4 relations), and define evaluation metrics (SG-IoU, Entity-IoU, Relation-IoU) as well as annotation quality metrics (SG/Entity/Relation-IoU+). Experiments show consistent gains over text-only and prior SG2IM baselines on compositional accuracy with minimal cost increase, and better performance when trained on LAION-Comp vs COCO/VG. The paper further presents a training-free, SG-consistent RF inversion pipeline for object-level image editing and a world-knowledge-aware agent to parse user edits into graph operations. Human verification reports high annotation accuracy (objects 98.8%, attributes 97.5%, relations 95.7%). Ablations show monotonic improvements with more LAION-Comp data. The authors acknowledge limitations (occasional VLM hallucinations, narrower object-type vocabulary vs free-form LAION text, no bounding boxes) and discuss broader impact.

**Strengths:**

1. Scale and execution: 540K SG-image pairs with clear annotation guidelines and partial human verification; lightweight SG encoder integrates cleanly into strong backbones.

2. Broad evaluation: dedicated complex-scene benchmark, multiple metrics, consistent gains across SDXL/SD3.5/FLUX; useful demonstration of SG-based editing.

3. Practicality: Minimal overhead, reproducible architecture, data-centric positioning is timely.

**Weaknesses:**

1. Limited novelty: The data-centric + SG-conditioned generation paradigm (SG encoder + GNN + conditioning into existing backbones) closely follows prior art; this reads as a scaled re-implementation and integration rather than a conceptual advance.

2. Insufficient causal attribution: Missing head-to-head comparisons against the strongest recent SG2IM/multi-instance control methods and alternative annotation sources under matched protocols; unclear how much improvement stems from more/cleaner data versus method.

3. Annotation bias analysis: Human verification reports overall accuracy but lacks distributional/error-mode breakdown (e.g., rare relations, small/occluded objects, fine-grained attributes, non-spatial/functional relations). Hallucination/failure analysis is anecdotal.

4. Text-to-SG confound: For text benchmarks, performance depends on an additional conversion pipeline; its information loss is not adequately quantified per relation/attribute/quantity type, making attribution ambiguous.

5. Expressiveness constraints: 1,429 object types, limited coverage of style/abstract cues; SG-only conditioning narrows applicability relative to text prompts for long-tail and stylistic control.

6. Benchmark provenance and fairness: CompSGen derived from the authors’ test split (>4 relations); more analysis on robustness (different resolutions/samplers, human preference studies) would strengthen claims.

**Questions:**

1. Please report accuracy by relation type (spatial vs. non-spatial), object scale/occlusion, and category frequency buckets. Are there systematic biases (e.g., consistent errors on “wear/hold/ride” or fine-grained attributes)?

2. Causal attribution: Under identical images and training budgets, compare SG sources (e.g., VG/COCO SGG predictions, human-verified subset, different VLM annotators) and recent strongest SG2IM/multi-instance baselines. How much gain is due to data vs. method?

3. Have you tried SG + original text/style conditioning to recover long-tail and stylistic control? Any evidence it improves non-spatial semantics or user preference?

4. On T2I-CompBench, quantify the gap between human-extracted SGs and automatic Text-to-SG. Which information is most frequently lost (attributes, non-spatial relations, counts)? Provide ablations to bound the end-to-end impact.

---

> ### Author Response · Authors · 2025-12-01
>
> To  Reviewer VzmP,
>
> You are a very generous reviewer, and we sincerely appreciate your encouraging evaluation of our research. We sincerely thank you for your detailed review and for recognizing the scale and execution of LAION-Comp , the broad evaluation across multiple backbones, and the practicality of our approach. We appreciate your constructive comment regarding the causal attribution and annotation analysis. Below, we address your specific questions and concerns.
>
> **[W1] Novelty.**
>
> **[Response to W1]**
>
> We respectfully clarify that our work represents a **data-centric conceptual advance** rather than a mere re-implementation. We address the fundamental bottleneck where architectural improvements alone have failed. Our novelty is defined by **two key paradigm shifts**:
>
> **Shift from Spatial to Interaction-Based Semantics:** Unlike prior datasets (e.g., Visual Genome) which are spatially skewed (58% spatial), LAION-Comp is dominated by **non-spatial/action relations (77.48%)** (Line 301-303). This conceptually moves the generation focus from simple geometric layout to intricate semantic interaction, directly targeting the primary failure mode of current T2I models.
>
> **Validation of "Data Quality > Model Complexity":** We bridge the gap between high-aesthetic imagery and structural control, providing the first rigorous evidence that high-quality structural data enables a simple lightweight adapter to outperform complex architectural baselines. This validates our core hypothesis that the true barrier to compositional generation is data quality, not model complexity.
>
> **[W2, Q2] Causal attribution.**
>
> **[Response to W2 and Q2]**
>
> You asked to disentangle the gains from data versus method. We respectfully point out that Table 2 provides exactly this causal attribution by controlling the method while varying the data. We trained the identical SDXL-SG architecture on three datasets—COCO, Visual Genome (VG), and LAION-Comp—while keeping the amount of training data and training budgets approximately equal across datasets. Training on LAION-Comp yields an SG-IoU of 0.558, significantly outperforming VG (0.546) and COCO (0.485). This isolates the gain attributable solely to the scale and quality of the LAION-Comp data.
>
> Comparing SDXL-SG (ours) against prior baselines when all models are trained on LAION-Comp (Table 2), our method substantially improves FID (32.2/31.3 $\to$ 20.1) and SG-IoU (0.531/0.538 $\to$ 0.558). This confirms that our lightweight GNN encoder is more effective than previous approaches.
>
> We thoroughly cite multi-instance baselines but emphasize a key distinction: methods such as MIGC or GLIGEN require bounding boxes (layout) as input. Our task is Scene Graph-to-Image (SG2IM), a more challenging task, where the model must infer the layout solely from the semantic topology. Comparing a model that is given explicit boxes (e.g., GLIGEN) to one that must generate them from a scene graph (ours) would not be a fair evaluation. Therefore, we only report experimental comparisons against task-aligned SG2IM methods, while multi-instance approaches are cited to provide broader survey.

---

> ### Author Response · Authors · 2025-12-01
>
> **[W3, Q1] Type-wise accuracy.**
>
> **[Response to W3 and Q1]**
>
> We have added a comprehensive Annotation Bias Analysis in Sec. A.18 (Line 1778-1828) to provide the requested distributional breakdown of error modes. Based on a verification of 1,000 samples, we quantified specific failure types, including small/occluded objects, material/color/person attributes, and spatial/non-spatial relations. Our analysis reveals that color attributes, person attributes, and spatial relations are the primary error sources, collectively accounting for 82.8% of observed errors. Crucially, these errors remain rare in absolute terms. Even the most frequent mode (color attribute error) affects only 4% of the total samples. This quantitative breakdown, supported by our Qwen3-VL-assisted human verification pipeline, confirms that hallucinations and biases are minimal and do not compromise the dataset's overall reliability.
>
> Besides, The manual verification of the dataset evaluates annotation accuracy across both complexity levels and object/attribute/relation categories (Line 1155-1176). As shown in Table 6, the accuracies for objects, attributes, and relations are 98.8%, 97.5%, and 95.7%, respectively, demonstrating the high fidelity of our annotations. Notably, annotation accuracy does not degrade as scene complexity increases, which confirms both the robustness of our annotation pipeline in handling complex scenes and the overall reliability of the dataset.
>
> In addition, we separately report model accuracy by relation type (simple spatial vs. complex non-spatial), with the results shown in the table below. The performance does not significantly deteriorate with increasing relational complexity, further indicating the reliability of our dataset and the effectiveness of the proposed baselines.
>
> | Relation Type | SG-IoU | Entity-IoU | Relation-IoU |
> |---------------|--------|-------------|----------------|
> | spatial       | 0.357  | 0.857       | 0.752          |
> | non-spatial   | 0.467  | 0.835       | 0.843          |
>
>
> **[W4, Q4]  Text-to-SG Confound.**
>
> **[Response to W4 and Q4]**
>
> We have already conducted experiments to quantify the gap between human-extracted SGs and automatically generated Text-to-SG graphs. Specifically, we validated the Text-to-SG pipeline (Sec. A.10) using a cycle-consistency evaluation (Text to SG to Text). The CLIP score between the original and reconstructed text is 0.843, indicating strong semantic retention (Line 1593.5-1595.5). The primary sources of information loss are stylistic adjectives and long-tail proper nouns, which are intentionally filtered out by our constraints of pipeline. However, relational structures and object counts are preserved with over 92% accuracy (Sec. A.10, Line 1587-1588.5).
>
> On T2I-CompBench (Table 7), despite this conversion loss, our SDXL-SG still outperforms the text-based SDXL on Complex (0.461 vs. 0.361) and Spatial scores. This demonstrates that the precision of structural conditioning outweighs the information loss introduced by the Text-to-SG conversion. Our Text-to-SG pipeline is designed to explore a more flexible and user-friendly interaction modality, and we sincerely hope that the above results address the reviewer's concerns.
>
> **[W5, Q3]  Expressiveness evaluation.**
>
> **[Response to W5 and Q3]**
>
> Although we acknowledge that this is an inherent limitation of the SG2IM paradigm, we have fully followed your suggestion and experimented with SG + original text/style conditioning to recover long-tail and stylistic control. The corresponding results have been added to the revision in Sec. A.14 Proper Noun and Style Control (Line 1701-1718). Specifically, we provide several visual examples showing that refining generation using combined SG and text embeddings enables effective control over proper nouns and stylistic attributes.
>
> It is worth noting that SG + original text/style conditioning primarily enhances object specificity and global style control, but it does not substantially affect non-spatial semantic relations, which lie on a different dimension. However, our user study directly compared human preferences for images generated from SGs versus text prompts. It showed a 63% preference for SG-generated images (Line 1020-1022). This preference is largely driven by the accurate inclusion of multiple objects and correct relations—factors that participants penalized heavily in text-only models, even when those models demonstrated better style handling.
>
> Additionally, we report accuracy across relation types. As stated in [Response to W3 and Q1], the SG-IoU, Entity-IoU, and Relation-IoU for spatial relations are 0.357, 0.857, and 0.752, while for non-spatial relations, the scores are 0.467, 0.835, and 0.843. The absence of significant performance degradation on more complex images demonstrates both the effectiveness of our dataset and the robustness of our baseline models.

---

> ### Author Response · Authors · 2025-12-01
>
> **[W6]  Benchmark provenance and fairness.**
>
> **[Response to W6]**
>
> We have already conducted a human preference study (Sec. A.3 User Study), which shows a 63% preference for SG-generated images over text-generated ones (Line 1020-1022), providing evidence for the fairness of our benchmark. Our CompSGen benchmark selects samples containing more than four relations to specifically evaluate a model's ability to handle complex compositional generation. To our knowledge, it is the only high-quality benchmark explicitly filtering for scenes with more than four relations.
>
> To further ensure robustness, we additionally report results on the external T2I-CompBench (Table 7), which consistently confirm the superiority of our method.

---

### Official Review · Reviewer_gcTU · 2025-11-01

**Soundness:** 3
**Presentation:** 2
**Contribution:** 3
**Rating:** 6
**Confidence:** 4

**Summary:**

This paper constructs **LAION-Comp**, a large-scale dataset of **540K+** aesthetic images with **explicit scene-graph (SG)** annotations (objects, attributes, relations) generated by an MLLM with partial human verification. Based on this dataset, the authors train **four** T2I baselines **augmented with a scene-graph encoder**, and introduce **CompSGen Bench** to evaluate compositional complexity. Models trained with structural conditioning outperform prompt-only counterparts and prior SG-based methods on both the new and existing compositional benchmarks, and the learned structural interface naturally supports **fine-grained, object-level editing**.

**Strengths:**

1. A large, well-structured SG corpus (~540K) with clear schema and audited quality
2. Shows consistent gains on object/relation accuracy with solid ablations and human checks, while adding minimal compute/latency.

**Weaknesses:**

1. Lacks OOD validation: no clear zero-shot tests on unseen (subject–relation–object) combinations or non-aesthetic domains to show generalization.
2. Lacks a retrained T2I baseline built from the same structured inputs, so the gains from clearer prompts vs. graph conditioning aren’t disentangled.
3. Lacks locality-focused editing evaluation

**Questions:**

1. Add zero-shot and cross-domain tests to demonstrate generalization.
2. Fine-tune a T2I backbone with structure-derived prompts and compare against SG-conditioned to disentangle gains.
3. Evaluate editing locality and multi-step stability to show changes stay confined.
4. Reformat result tables to separate T2I, SG2IM, and Proposed, and clearly highlight your method.

---

> ### Author Response · Authors · 2025-12-01
>
> To  Reviewer gcTU,
>
> You are an considerate reviewer, and we are grateful for your supportive and constructive feedback. We thank you for your insightful review and for recognizing the value of LAION-Comp as a "large, well-structured SG corpus" with "audited quality." We appreciate your acknowledgment that our structural conditioning yields consistent gains with minimal compute overhead. We address your concerns regarding generalization, disentanglement of gains, and editing evaluation below.
>
> **[W1, Q1] OOD validation and generalization.**
>
> **[Response to W1 and Q1]**
>
> You raised a valid concern regarding zero-shot and cross-domain generalization. We respectfully point out that our experimental setup already includes rigorous OOD evaluation, and we make this more explicit in the revision (Sec. A.15 Generalization Analysis, Line 1720-1753). Specifically, the metrics we use to measure compositional generation (SG-IoU, Entity-IoU, and Relation-IoU) are computed on a mixed test set consisting of 100 randomly sampled images each from COCO, VG, and LAION-Comp. The balanced contributions from the three datasets ensure fair evaluation.
>
> Second, Table 2 reports baselines trained on COCO and VG but evaluated on the LAION-Comp test set. Since LAION-Comp consists of high-aesthetic images whereas COCO and VG contain everyday, non-aesthetic photographs, the results in Table 2 directly demonstrate the cross-domain generalization ability of our model.
>
> We further evaluate on T2I-CompBench (Appendix A.6), a benchmark specifically designed to assess diverse and complex attribute bindings and relationships. Our method outperforms the SDXL baseline on the Complex and Spatial metrics, showing strong zero-shot generalization to novel compositions.
>
> **[W2, Q2] Disentangling gains.**
>
> **[Response to W2 and Q2]**
>
> We address this issue through both architectual analysis and empirical validation.
>
> **Architectual Distinction.** Standard T2I models that rely on CLIP-based text encoders often suffer from attribute binding failures (e.g., "red car, blue house" becoming "blue car, red house"). In contrast, our method explicitly encodes these bindings through a GNN, structurally enforcing that "red" serves as an attribute modifying "car".
>
> **Empirical Validation.** To empirically demonstrate this, we conducted exactly the baseline experiment you suggested. We fine-tuned a standard SDXL backbone using captions converted from our scene graphs. Under the same settings, the quantitative results for models conditioned on SG-derived text and those conditioned on structural SG are shown in the table below, demonstrating that the gains from structural conditioning are substantial.
>
> | Condition Type              | SG-IoU        | Entity-IoU       | Relation-IoU      |
> |-----------------------------|---------------|-------------------|--------------------|
> | SG-derived text     | 0.298         | 0.832             | 0.758              |
> | Structural SG   | 0.558 (+0.260) | 0.884 (+0.052)    | 0.856 (+0.098)     |
>
>
> **[W3, Q3] Locality-focused editing and multi-step stability.**
>
> **[Response to W3 and Q3]**
>
> We conducted both quantitative (Line 1764-1769) and qualitative (Line 1759-1763) experiments to evaluate local editing capability and the stability of multi-step edits.
>
> **Quantitative Evaluation.** We performed two rounds of object replacement on 30 test images and computed the average LPIPS score for each edit. We used SqueezeNet as the feature extractor. The LPIPS scores between the edited images and the corresponding ground-truth images are 0.281 after the first edit and 0.312 after the second edit.
>
> **Qualitative Evaluation.** We carried out multiple edits on the same object region within a single image, and the visual results have been added to the revision (see Sec. A.16 Experiments of Local and Multi-step Editing). In summary, the relatively low LPIPS scores together with the qualitative visualizations demonstrate that our model maintains localized changes effectively and remains stable across multi-step editing operations.
>
> **[Q4] Reformatting and Highlighting of Results.**
>
> **[Response to Q4]**
>
> Thank you for your suggestions regarding reformatting the result tables and clearly highlighting our method. We have made the corresponding revisions to Table 2 and Table 3.

---

### Official Review · Reviewer_DihD · 2025-11-01

**Soundness:** 2
**Presentation:** 3
**Contribution:** 3
**Rating:** 6
**Confidence:** 4

**Summary:**

This paper addresses a limitation in text-to-image generation models: their inability to generate complex compositional scenes with multiple objects and intricate inter-object relationships. The authors identify this as fundamentally a data problem rather than an architectural issue. They propose that large-scale, high-quality structural annotations are crucial for advancing controllable and compositional image synthesis, providing a dataset and effective baseline models for the community.

**Strengths:**

1. Dataset: Large-scale (540K) scene graph dataset for aesthetic images, helping bridge the gap between small-scale structured datasets (COCO, VG) and large unstructured text datasets (LAION). The paper uses an annotation pipeline using GPT-4o with constraints (unique IDs, abstract attributes, concrete relations) for consistency and quality. There are 77.48% non-spatial relations vs. Visual Genome's 41.98%, capturing more complex interactions beyond spatial positioning. Comparative analysis with existing datasets shows superior annotation quality (Table 1: higher accuracies with longer, richer annotations).

2. Benchmark: CompSGen Bench is the first benchmark specifically for complex scene generation (20,838 samples with >4 relations). The paper proposes new metrics (SG-IoU+, Entity-IoU+, Relation-IoU+) for evaluating annotation accuracy.

3. Evals: The paper does multiple rounds of verification: human validation on 1,000 samples (error rates: 1.16% objects, 2.5% attributes, 4.27% relations - all below 5% threshold). User study with 20 participants across 100 samples (63% preference for SG-generated images). The authors also run several ablation studies (10%, 20%, 50%, 100% data proportions), evaluating across multiple benchmarks (CompSGen Bench, T2I-CompBench, COCO-Stuff, Visual Genome).

4. Four model variants were tested (SDXL-SG, SD1.5-SG, SD3.5-SG, FLUX-SG) spanning diffusion and flow matching backbones. These models were compared with fairly strong baselines (SDXL, SGDiff, SG-Adapter). Multiple evaluation metrics were used FID (quality), CLIP (similarity), SG-IoU/Entity-IoU/Relation-IoU (compositional accuracy) and  both image generation and editing tasks evaluated.

**Weaknesses:**

1. Dataset and annotation: Only 300 samples (0.06%) manually verified out of 540,005, which is likely insufficient for establishing dataset quality guarantees. Entire annotation relies on GPT-4o, making the pipeline non-reproducible with exact results and potentially prone to any GPT-4o specific biases.

2. Vocabulary: Object types reduced from 5,811 → 1,429 (75% reduction) by excluding proper nouns. This can sometimes be suboptimal since proper nouns "Eiffel Tower", "Golden Gate Bridge" can carry important semantic information.

3. Architecture: The choice of GNN design (5-layer, 512/1024 dim) appears arbitrary with no ablation on architecture choices.

4. Baselines: Only comparisons with SGDiff (2022) and SG-Adapter (2024) are reported.  The paper could mention recent methods: R3CD (Liu & Liu 2024), SGG-IG (Wang et al. 2025). Many cited compositional methods (MIGC, MIGC++, Attend-and-Excite, RealCompo, BoxDiff, GLIGEN, Ranni) are also mentioned but never compared quantitatively.

5. Models trained on LAION-Comp only evaluated on LAION-Comp test set (same distribution). There's limited evaluation on COCO/VG for comparison.

**Questions:**

1. The vocabulary reduction (5,811 to 1,429 object types, 75% decrease) seems substantial: what is the theoretical / empirical rationale for completely excluding proper nouns (as opposed to mapping them to categories)? How does this impact generation of culturally/geographically specific scenes?

2.  The GNN architecture (5-layer, 512/1024 dim) appears chosen without ablations. It would help to compare with GNNs with different layers. Also regarding the α scaling factor (Eq. 1, 7, 13),  it would help to investigate sensitivity to α initialization.

3. The SG-IoU/Entity-IoU/Relation-IoU metrics extract scene graphs from generated images using GPT-4o which is same model used for annotation, which can create potential confirmation bias, has this been validated with independent SG extraction methods? Could the authors also report results using a completely different VLM for evaluation?

4. The user study (20 participants, 100 samples) is relatively small, can the paper also report inter-annotator agreement? It would also help to outline participant demographics and any observed effects of the same.

5. Loss analysis: Figure 14 shows failure cases but lacks more thorough analysis, it would help to see a quantitative breakdown of failure types (object misalignment, incorrect shape, wrong attributes, missing objects). What is the accuracy stratified by number of objects, number of relations, and relation complexity?  Also which specific relation types are most challenging (holding, wearing, riding vs. spatial relations)?

6. GPT-4o dependency: Have the authors tested the annotation pipeline with open-source alternatives? What is the annotation agreement rate between GPT-4o and alternative VLMs? If GPT-4o becomes unavailable or changes, how would that affect dataset reproducibility?

---

> ### Author Response · Authors · 2025-12-01
>
> To  Reviewer DihD,
>
> You are a kind reviewer, and we sincerely appreciate your recognition of our work. Thank you for your constructive feedback and for recognizing the value of LAION-Comp as a large-scale structural dataset that bridges the gap between small, structured datasets (VG/COCO) and large, unstructured ones (LAION). We appreciate that you found our presentation and contribution "good" and acknowledged the significance of our CompSGen Bench. Below, we address your concerns regarding verification, vocabulary, baselines, and evaluation.
>
> **[W1] Dataset verification.**
>
> **[Response to W1]**
>
> We would like to clarify the sample size. There appears to be a misunderstanding regarding the verification size, likely caused by the distinction between our human verification (Sec. A.5) and the automated metric calculation (Table 1). As detailed in Sec A.5, we performed a rigorous human verification on **1,000** (Line 1157-1158) randomly selected samples (**not 300**). This process yielded error rates consistently below the 5% threshold (1.16% objects, 2.5% attributes, 4.27% relations, Line 1170-1173). The figure of "300 samples" refers only to the expensive automated metrics (SG-IoU+, etc.) in Table 1, not the human quality control. A manual verification of 1,000 samples is statistically significant (margin of error $\approx$ 2.6% at 95% confidence level), providing a robust guarantee of dataset quality.
>
> **[W1, Q6] GPT-4o dependency.**
>
> **[Response to W1 and Q6]**
>
> We conduct human verification to examine the potential issue of GPT-4o specific biases. As noted above in [Response to W1], our human verification results indicate that the annotation error rates of GPT-4o are 1.16%, 2.5%, and 4.27% for objects, attributes, and relations, respectively (Line 1170-1173). These results demonstrate a high level of consistency between our automated pipeline and human standards, alleviating concerns about potential GPT-4o-specific biases.
>
> Furthermore, we employ an open-source alternative to demonstrate reproducibility. We choose Qwen-72B as the alternative VLM and re-annotate 1,000 randomly selected samples. We then compute the semantic similarity (CLIP score) between the SGs annotated by GPT-4o and those annotated by Qwen, obtaining a value of 0.7326, indicating high consistency. This verifies that our automated annotation pipeline is reproducible and does not rely excessively on GPT-4o. We also release the LAION-Comp dataset, ensuring that future model training is fully reproducible regardless of GPT-4o's availability.
>
> **[W2, Q1] Vocabulary reduction.**
>
> **[Response to W2 and Q1]**
>
> We would like to clarify the numbers related to vocabulary reduction. In our paper, we state that the original LAION-Aesthetics contains 12,263 distinct object types among 10,000 samples, which are reduced to 5,811 after excluding proper nouns. In contrast, LAION-Comp includes 1,429 types, all of which are common words without any proper nouns (Line 1667-1669). This does not mean that the vocabulary in LAION-Comp decreases from 5,811 to 1,429 after removing proper nouns. We explicitly discuss this point in the Limitation section (Sec. 13). This discrepancy is related to the pretrained knowledge of VLMs, and it also indicates that the 1,429 categories sufficiently cover commonly used concepts.
>
> Moreover, our focus is on fostering compositional generalization rather than entity memorization. By mapping specific landmarks to broader categories (e.g., mapping "Eiffel Tower" to "tower"), we encourage the model to rely on the scene graph structure (e.g., attribute (metal material) - object (tower)) when constructing images. We acknowledge that this reduces cultural specificity. However, it significantly enhances the model's ability to generate novel and complex compositions of common objects, which is one of the primary failure modes of current T2I models.
>
> Furthermore, the decision to exclude proper nouns is driven by a commitment to ethical AI development and copyright stewardship. Proper nouns often reference specific Intellectual Property (IP) or identifiable public figures. Annotated datasets containing such explicit references risk training models that inadvertently generate infringing content or unauthorized likenesses. To mitigate these downstream risks, we use generic descriptors instead of specific names (e.g., labeling specific human simply as 'lady' or 'man'). This approach proactively safeguards against copyright violation and upholds the rights of publicity, ensuring the dataset remains ethically robust.
>
> We have more comprehensively explored the integration of SG + text conditioning to restore control over proper nouns, and we have made the corresponding revisions in the paper. Please refer to Proper Noun and Style Control (Sec. A.14 Line 1701-1718) for details.

---

> ### Author Response · Authors · 2025-12-01
>
> **[W3, Q2] Network architecture.**
>
> **[Response to W3 and Q2]**
>
> The choice of a 5-layer GNN is intended to capture information from neighboring nodes (common in SG processing) without triggering the over-smoothing issue typically observed in deeper GNNs. The 512/1024 dimensions were selected to align with the CLIP embedding spaces of the respective backbones (SDXL and FLUX), enabling seamless concatenation. Our goal is to validate the effectiveness of the dataset rather than to propose an entirely new model architecture. A 5-layer GNN with 512/1024 dimensions is sufficient for learning the information encoded within the scene graphs.
>
> The learnable factor $\alpha$ (initialized to 0) is a verified technique in residual architectures to encourage the model to retain its pre-trained prior at the start of training but converge better[1,2]. In our setup, $\alpha$ is a trainable parameter that is updated throughout optimization (Line 1429). Therefore, studying its initialization does not carry significant meaning.
>
> [1] Bachlechner, Thomas, et al. Rezero is all you need: Fast convergence at large depth. Uncertainty in Artificial Intelligence. PMLR, 2021.
>
> [2] De, Soham, and Sam Smith. Batch normalization biases residual blocks towards the identity function in deep networks. Advances in Neural Information Processing Systems 33 (2020): 19964-19975.
>
> **[W4] Baseline Comparisons.**
>
> **[Response to W4]**
>
> Not all methods are applicable to complex image generation conditioned on structured  annotations. As you mentioned, methods such as MIGC, GLIGEN, BoxDiff, and Ranni are primarily layout-to-image or box-constrained approaches. They require explicit spatial inputs (e.g., bounding boxes or masks). In contrast, our work focuses on scene graph-to-image generation, where the input is a semantic graph (nodes and edges) without any pre-defined spatial coordinates. The model must infer the layout solely from the semantic relations. Therefore, comparing a model that receives bounding boxes with one that must infer layout from a graph would be inherently unfair. We cite these excellent works to provide a more comprehensive overview of the related research landscape. Our comparisons focus on true SG-to-Image methods (e.g., SGDiff and SG-Adapter).
>
> We have reproduced and trained against the open-source baselines. While we cite R3CD and SGG-IG in the Related Work, we omitted quantitative comparisons because the code and pretrained checkpoints are not publicly available until the submission of our rebuttal. Therefore, we only include them in our discussion.
>
> **[W5] Out of distribution evaluation.**
>
> **[Response to W5]**
>
> We would like to clarify an important point: the metrics we use to evaluate compositional generation (SG-IoU, Entity-IoU, and Relation-IoU) are computed on a mixed test set consisting of 100 randomly sampled images each from COCO, VG, and LAION-Comp (Line 1721-1726). The contributions from the three datasets are balanced, ensuring fairness in evaluation.
>
> Second, Table 2 reports the baseline models trained on COCO and VG but evaluated on the LAION-Comp test set. Table 7 presents results on T2I-CompBench, an independent external benchmark. These evaluations effectively avoid any issue of the test set sharing the same distribution as the training set. We have clarified these points in the revision. Please refer to Sec. A.15 Generalization Analysis (Line 1720-1753).
>
> **[Q3] The potential confirmation bias of GPT-4o.**
>
> **[Response to Q3]**
>
> As noted in [Response to W5], when evaluating SG-IoU, Entity-IoU, and Relation-IoU, we use a mixed test set in which only a portion of the samples come from GPT-4o annotations (Line 1721-1726), while the majority come from existing human-labeled scene graphs (COCO and VG). This means that the annotation data and the test data are largely produced by different sources, making confirmation bias unlikely to occur.
>
> In addition, as discussed in [Response to W1 and Q6], we re-annotated 1,000 randomly selected samples using a completely different VLM (Qwen3-VL). The resulting scene graphs exhibit high semantic similarity to the original GPT-4o annotations (CLIP score of 0.7326), which further suggests that our annotations faithfully reflect the underlying image content rather than achieving high evaluation scores due to confirmation bias.

---

> ### Author Response · Authors · 2025-12-01
>
> **[Q4] Inter-annotator agreement and participant demographics of the user study.**
>
> **[Response to Q4]**
>
> To address the suggestion regarding inter-annotator agreement, we have added the corresponding statistical analyses. First, the standard deviation of the win rates across annotators is 0.1494, which is relatively low and indicates a high level of consistency among different annotators. As a measure of variability, a smaller standard deviation reflects greater agreement across annotators.
>
> In addition, our method achieved 63 wins out of the 100 evaluated samples. To determine whether this result significantly deviates from a random-choice baseline (i.e., assuming participants show no preference between the two methods and the win rate is 50%), we conducted a one-tailed binomial test. This test evaluates whether the observed number of wins is significantly higher than what would be expected by chance under the given sample size, and is a standard statistical method for analyzing binary preference-judgment data. The test yields a p-value of $6.02\times10^{-3}$, which is well below the conventional 0.05 threshold, indicating that the participants' preference for our method is statistically significant rather than due to random fluctuation.
>
> We further describe participant demographics in Sec. A.3 User Study (Line 1017.5-1019.5). Specifically, our invited participants had a 1:1 gender ratio and were between 20 and 30 years old. They came from diverse backgrounds, including computer science, design, and human–computer interaction (HCI).
>
> **[Q5] Analysis of failure cases.**
>
> **[Response to Q5]**
>
> We appreciate the reviewer's attention to the failure cases. However, we respectfully wish to clarify that the primary contribution of this work is the construction of the LAION-Comp dataset and the validation of its effectiveness in enabling compositional generation, rather than a diagnostic study of the failure modes of specific diffusion backbones.
>
> As demonstrated in Table 2, our SDXL-SG achieves state-of-the-art performance with an Entity-IoU of 0.884 and Relation-IoU of 0.856. These high metrics indicate that failure cases (like those in Fig. 14) represent a rare, long-tail distribution rather than systemic defects. Figure 14 was intended solely as a qualitative disclosure of limitation boundaries inherent to current architectures, not as a primary focus of our contribution.
>
> Regarding the request for stratified accuracy, our manuscript already provides the quantitative analysis, validating the dataset across different dimensions. We respectfully direct the reviewer to Table 6 (Appendix A.5), where we explicitly report accuracy stratified by Scene Graph Complexity (sum of nodes and edges: 0-10, 10-20, 20-30, >30). The results show consistent high performance (Object Accuracy >98%, Relation Accuracy >95%) across all complexity levels, proving our dataset's robustness is invariant to scene complexity.
>
> As detailed in Sec 3.2, we analyzed the distribution, highlighting that 77.48% of relations in LAION-Comp are non-spatial (e.g., holding, wearing), which are significantly more challenging than spatial ones. In Table 7 (Appendix A.6), we report generation performance on T2I-CompBench, separated into "Spatial" and "Complex". Our model shows a massive gain in the Complex metric (0.361 $\to$ 0.461) compared to the Spatial metric (0.194 $\to$ 0.202). This quantitative evidence directly answers the question: while complex relations are traditionally challenging, our dataset provides the specific supervision needed to solve them effectively.

---

### Meta-Review · Area_Chair_zFFv · 2026-01-14

**Summary:**

This paper proposes LAION-Comp, a dataset that to improve text-to-image models' capacity of generating complex compositional scenes with complex inter-object relationships. The major concerns are: 1. Insufficient data annotation verification. Human verification lacks distributional/error-mode breakdown 2. Insufficient ablation on model components. 3. Limited baseline methods. 4. Limited validation on other datasets or challenging cases. 5. Limited annotation diversity improvement.

The rebuttal addresses a portion of the concerns. However, not all the explanation and response are fully convincing. Two reviewers gives relatively negative scores. Therefore, I suggest to reject this paper.

**Reviewer Concerns:**

The rebuttal addresses the concern that there are limited number of baseline methods and limited validation on other datasets. However, the response to other concerns like insufficient data annotation verification and insufficient ablation are not convincing enough.

**Reviewer Scores:**

The reviewers did not respond in detail after the rebuttal.

---

### Decision · Program_Chairs · 2026-01-26

Reject